# RNA G-quadruplexes control mitochondria-localized mRNA translation and energy metabolism

Leïla Dumas[1,5], Sauyeun Shin[1,5], Quentin Rigaud[1], Marie Cargnello[1], Beatriz Hernández-Suárez [1], Pauline Herviou [1], Nathalie Saint-Laurent[1], Marjorie Leduc [2], Morgane Le Gall [2], David Monchaud [3], Erik Dassi [4] ✉, Anne Cammas [1] ✉ & Stefania Millevoi [1] ✉

Cancer cells rely on mitochondria for their bioenergetic supply and macro-molecule synthesis. Central to mitochondrial function is the regulation of mitochondrial protein synthesis, which primarily depends on the cytoplasmic translation of nuclear-encoded mitochondrial mRNAs whose protein products are imported into mitochondria. Despite the growing evidence that mitochondrial protein synthesis contributes to the onset and progression of cancer, and can thus offer new opportunities for cancer therapy, knowledge of the underlying molecular mechanisms remains limited. Here, we show that RNA G-quadruplexes (RG4s) regulate mitochondrial function by modulating cytoplasmic mRNA translation of nuclear-encoded mitochondrial proteins. Our data support a model whereby the RG4 folding dynamics, under the control of oncogenic signaling and modulated by small molecule ligands or RG4-binding proteins, modifies mitochondria-localized cytoplasmic protein synthesis. Ultimately, this impairs mitochondrial functions, affecting energy metabolism and consequently cancer cell proliferation.

Contrary to conventional wisdom[1], accumulating evidence supports the notion that mitochondrial metabolism is active in cancer cells and required for tumor growth, as it fuels the bioenergetic and biosynthetic needs of cancer cells[2]. Metabolic reprogramming with increased dependence on mitochondrial functions may increase during tumor progression[3] and in response to therapies[4,5]. Central to the mitochondrial function is the regulation of mitochondrial protein synthesis, which relies primarily on the cytoplasmic translation of nuclear-encoded mitochondrial mRNAs whose protein products are then imported into mitochondria[6]. The majority of these proteins are directly or indirectly involved in the mitochondrial translation of 13 mitochondrial-encoded membrane proteins.

These proteins are all components of the respiratory chain, which generates the majority of cellular ATP via oxidative phosphorylation (OXPHOS). Recent data have shown that most mitochondrial protein-coding mRNAs are enriched at the mitochondrial outer membrane (OMM) in a translation-dependent manner and translated by mitochondria-associated cytoplasmic ribosomes, supporting the notion of localized translation at the OMM regulating the influx of mitochondrial proteins[7]. Despite the growing awareness that mitochondrial protein synthesis contributes to the onset and progression of cancer and thus offers new opportunities for cancer therapy[8–10], knowledge about the underlying molecular mechanisms remains limited.

[1]Centre de Recherches en Cancérologie de Toulouse (CRCT), Université de Toulouse, Equipe Labellisée Fondation ARC, Université de Toulouse, Inserm, CNRS, Université Toulouse III-Paul Sabatier, Toulouse, France. [2]Proteom'IC facility, Université Paris Cité, CNRS, INSERM Institut Cochin, Paris, France. [3]Institut de Chimie Moléculaire (ICMUB), UBFC Dijon CNRS UMR6302, Dijon, France. [4]Laboratory of RNA Regulatory Networks, Department of Cellular, Computational and Integrative Biology (CIBIO), University of Trento, Trento, TN, Italy. [5]These authors contributed equally: Leïla Dumas, Sauyeun Shin. ✉e-mail: erik.dassi@unitn.it; anne.cammas@inserm.fr; stefania.millevoi@inserm.fr

mRNA translation is the most energetically consuming process in the cell and its deregulation is widely recognized as a cancer hallmark[11,12]. Current knowledge points towards the concept of mRNA translational reprogramming that selectively controls subgroups of mRNAs (also called "regulons"[13]) involved in defined cancer pathways[14]. Leveraging this concept to target specific oncogenic pathways for therapeutic purposes is currently limited by the lack of in-depth knowledge of the molecular determinants, i.e. RNA elements/structures and RNA-binding proteins (RBPs), as well as their mode of action and regulation, which, according to recent data, may involve specific localized factories[15]. RNA G-quadruplexes (G4), hereafter referred to as RG4s, are non-canonical structures consisting of G-rich sequences that form stacked tetrads of guanines (G-quartets) held together by Hoogsteen hydrogen bonds (for recent reviews, see Dumas et al. and Varshney et al.[16,17]). The pervasive nature of RG4s has been demonstrated by surveying the extent of their folding both in vitro and in cellulo using sequencing-based methods[18,19] and optical imaging via immunodetection with G4-specific antibodies (including BG4[20]) or molecular probes (including biomimetic fluorescent probes[21]). The transcriptome-wide analysis of RG4 formation in living cells indicate that these are transient structures whose folding is regulated by RBPs and helicases[18,22], many of which have been identified by large-scale proteomic studies[23–31]. Although RG4s are increasingly recognized as important in the translational regulation of cancer protein-coding mRNAs, their ability to target mRNA regulons was shown only by few studies[26]. Recent data highlighting the role of cytoplasmic translation in mitochondrial protein synthesis, suggested a potential role for RG4s in cytoplasmic translation of nuclear-encoded mRNAs[32]. This is consistent with the emerging view that G4s play an important role in mitochondria[16,33]. Supporting this notion, G4s are found in mitochondria[34–36] where they function as modulators of genome replication, transcription[37], and RNA stability[38]. Although this indicates a role for DNA/RNA G4s in mitochondrial biogenesis and mitochondrial DNA (mtDNA) gene expression, the possibility that RG4s affect mitochondrial functions by regulating cytoplasmic expression of mitochondrial proteins remains yet unconfirmed.

Here we show that cytosolic RG4s regulate mitochondrial function by controlling cytoplasmic translation of nuclear-encoded mitochondrial mRNAs at the outer mitochondrial membrane. This function, which is mediated by their interaction with the RNA-binding protein hnRNP U and is controlled by mTOR oncogene signaling, results in altered mitochondrial functions affecting energy metabolism and therefore cancer cell proliferation.

## Results

### RG4s affect mitochondrial gene expression and function

To investigate the link between RG4s and mitochondria, we first performed a functional enrichment analysis using transcriptome-wide datasets of in silico predicted[39] or in vitro (HeLa or HEK293T cells) identified RG4s[18,19]. This analysis revealed that several mitochondria-related terms were significantly enriched in RG4-containing mRNAs (Supplementary Fig. 1a). Consistent with this and with the notion that RG4s modulate gene expression[16], we found that mitochondrial proteins encoded by RG4-containing mRNAs, including factors of the respiratory chain complex, were significantly enriched among differentially expressed proteins after treatment of HEK293T or HeLa cells with the RG4 ligand carboxypyridostatin (cPDS, a chemical derivative of PDS)[40] (Supplementary Fig. 1b). These results, together with previous data highlighting oxidative phosphorylation as the most significantly enriched pathway among the proteins differentially expressed in HeLa cells treated with the G4 ligand 20A[41], suggest that RG4s may be involved in the regulation of mitochondrial gene expression and respiratory function in a cell type- and ligand-independent manner. To address these possibilities, we reasoned

that RG4s regulating mitochondrial mRNA expression should be found in close proximity to mitochondria. Previous works using fluorescent G4 ligands or the BG4 antibody[34–36] mapped G4s to the mitochondrial genome. This result is consistent with the presence of G-rich sequences in mitochondrial DNA, but due to their uneven distribution between the two strands, the resulting transcriptome will have high or low G content. Indeed, in humans, the heavy DNA strand is G-rich and serves as a template for the transcription of most mitochondria-encoded genes, giving rise to RNAs (mRNAs, rRNAs, tRNAs) with low G-content. The light DNA strand is instead G-poor and mostly transcribes noncoding RNAs, which form RG4 structures and are degraded by the degradosome[38]. Based on these notions, it is therefore expected that mitochondria form DNA G4s but very little RNA G4s. In agreement with previous findings[34–36], we observed that the BioCyTASQ biomimetic probe, previously shown to highlight cytoplasmic RG4s foci[21], colocalized with the OMM protein TOMM20 (Fig. 1a, b) or the mitochondrial marker (MitoTracker Deep Red) in HeLa cells (Supplementary Fig. 1c) but was lost after RNAse treatment (Supplementary Fig. 1d). This result together with the observation that TOMM20 did not colocalize with a non-mitochondrial cytosolic protein (Supplementary Fig. 1e, f) suggests that RG4s probes mapped RG4s at mitochondria. As expected, the BioCyTASQ signal increased after addition of the G4-stabilizing ligand PhenDC3 (Fig. 1a, b, Supplementary Fig. 1c, g). However, this was accompanied by a change in colocalization with mitochondria due to the spreading of the RG4 signal toward the cytosolic area (Fig. 1a, b, Supplementary Fig. 1c). Overall, these data suggest that some cytosolic RG4s mapped to mitochondria and their stabilization increased RG4 formation, which is accompanied by a modification in their proportional co-distribution. Next, we asked if PhenDC3 impacted mitochondrial functions and biogenesis in three cancer cell lines (HeLa, HCT116 and SKMEL28). In HCT116 cells, we found that PhenDC3 reduced the oxygen consumption rate (OCR) as measured by Seahorse assays (Fig. 1c) and the membrane potential (Fig. 1d), without affecting cell viability and number after 24 h treatment (Supplementary Fig. 2a). In agreement with previous findings[35], PhenDC3 did not modify mitochondrial mass (Supplementary Fig. 2b). Consistent with the emerging notion that proliferation depends on mitochondrial activity[42,43], we observed that PhenDC3 inhibited proliferation of the three cell lines from day 4 with effects whose magnitude is correlated with mitochondrial respiratory function (Fig. 1c, e, Supplementary Fig. 2c, d). Together, these results indicate that RG4s are important cis-elements regulating the cytoplasmic expression of nuclear-encoded mitochondrial mRNAs and that modulation of their conformation affects both mitochondrial activity and cancer cell proliferation.

### RG4s are molecular determinants of OMM-localized translation

Since nuclear-encoded mitochondrial mRNAs could partly localize and undergo translational regulation at the outer membrane of mitochondria (OMM)[15,44,45], we analyzed whether RG4-containing mRNAs exhibit a specific spatial distribution using data from APEX-seq, a method based on direct proximity labeling of RNA using the peroxidase enzyme APEX[7]. We found that RG4-containing mRNAs were more enriched at the OMM compared to the ERM (Endoplasmic Reticulum Membrane) (Supplementary Fig. 3a). Using RNA immunoprecipitation (RIP) assays with HCT116 cytoplasmic extracts and the BG4 antibody[20], we validated that a subset of these OMM-associated mRNAs (for which RG4s were identified in cellulo[18] and/or in vitro[19]) was prone to form RG4s in cellulo (Fig. 2a, Supplementary Fig. 3b). We found that PhenDC3 modified the protein expression but not the levels of these RG4-containing mRNAs (Fig. 2b, c and Supplementary Fig. 3c), suggesting that RG4 stabilization affects the translation of OMM-associated mRNAs. Finally, given that mitochondrial translation depends on the coordination between the cytoplasmic and the mitochondrial machinery[46], we tested whether the deregulation of

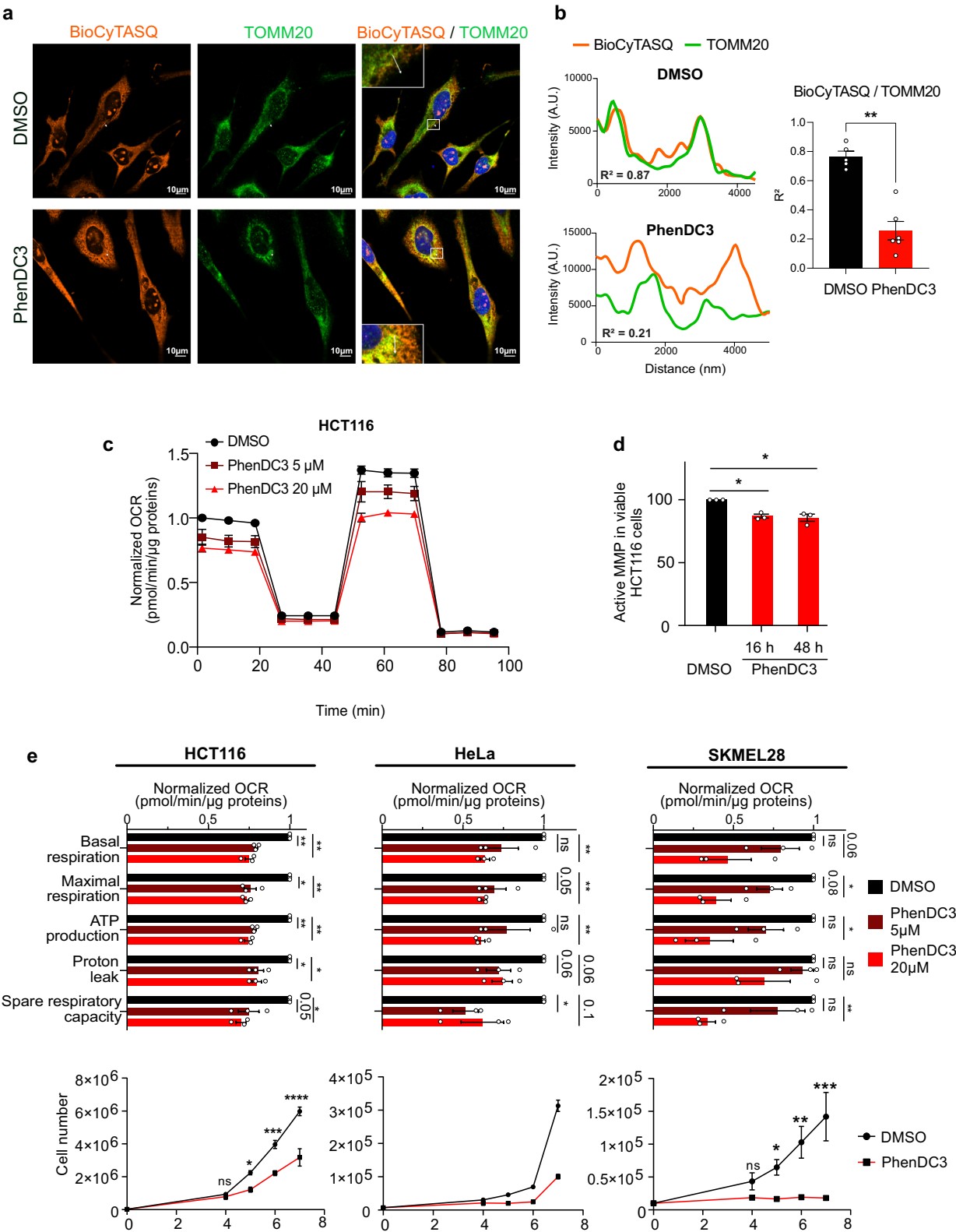

RG4-dependent cytoplasmic translation could have an impact on the expression of proteins encoded by the mitochondrial genome via RG4-less mRNAs and translated by mitoribosomes. We found that PhenDC3 preferentially decreased COX-1 as compared to COX-4 (Supplementary Fig. 3d), which are mitochondria-encoded and nuclear-encoded subunits of the complex IV in the electron transport chain in mitochondria, respectively.

The prevailing consensus on nuclear-encoded mitochondrial mRNA expression is that their translation, often relying on 3'UTR sequences[47], is accomplished by OMM-associated ribosomes[48]. Their recruitment at this specific location may depend on the assembly of the translation machinery (ribosome-dependent mechanism) or rely on specialized RBPs responsible for localizing them in the vicinity of the mitochondria (ribosome-independent mechanism)[7].

**Fig. 1 | RG4 localization and function in mitochondria. a** RG4 detection in fixed cells using the BioCyTASQ probe (1 μM for 24 h) coupled with TOMM20 immuno-fluorescence in HeLa cells treated with DMSO or 20 μM of PhenDC3 for 24 h. Scale bar: 10 μm. Shown is a representative result from n = 3 independent experiments. **b** Graphs displaying the fluorescence intensity (arbitrary unit) in each channel over the distance depicted by the white arrow in (**a**). The correlation coefficient ($R^2$) of the colocalized fluorescence intensities of BioCyTASQ with TOMM20 signal is plotted. Data are presented as mean values ± SEM for 5 or 6 different cells, *P*-value = 0.0024. **c** Quantification of the normalized Oxygen Consumption Rate (OCR) per cell, measured by a Seahorse assay, in HCT116 cells treated with DMSO or a dose scale of PhenDC3 for 16 h. Data are presented as mean values ± SEM of n = 3 independent experiments. **d** Quantification of the Mitochondrial Membrane Potential (MMP) using flow cytometry analysis of the fluorescent TMRE probe in

HCT116 cells treated with DMSO or PhenDC3 for 16 h or 48 h and plotted relatively to the DMSO condition. Data are presented as mean values ± SEM of n = 3 independent experiments, *P*-value = 0.0135 and *P*-value = 0.0373 for 16 h and 48 h PhenDC3 treatment, respectively. **e** Quantification of the Oxygen Consumption Rate (OCR) linked to basal and maximal mitochondrial respiration, ATP turnover, proton leak, spare respiratory capacity, measured by a Seahorse assay, and of the proliferation in HCT116, HeLa and SKMEL28 cells treated with DMSO or a dose scale of PhenDC3 for 16 h. Data are presented as mean values ± SEM of n = 2 (for pro-liferation assay in HeLa) or n = 3 independent experiments. For all the panels, *P < 0.05, **P < 0.01, ***P < 0.001, ns: non-significant (two-sided paired t-test for OCR and two-way ANOVA for the proliferation). For (**b**, **c**, **d**, **e**) Source data are provided as a Source Data file indicating exact *P*-values for (**e**).

Nuclear-encoded RG4-containing mitochondrial mRNAs showed higher enrichment in the ribosome-dependent group (Supplementary Fig. 3e), with a predominant location of the RG4s in the 3′UTR (Supplementary Fig. 3f). To define whether RG4s are molecular determinants of OMM-localized translation, we started by monitoring an endogenous ribosome-dependent mRNA. We focused on AKAP1, a scaffolding protein that acts as a mitochondrial signaling hub by recruiting protein kinase A (PKA) to the OMM and regulates mito-chondrial biogenesis, oxidative metabolism, and cell survival[49]. Indeed, AKAP1 mRNA was found to colocalize with its encoded protein at mitochondria in a translation-dependent manner, suggesting that this mRNA could be translated locally[15]. Specifically, AKAP1 mRNA was delocalized and lost its colocalization with the encoded protein after treatment with puromycin, a translation inhibitor[15]. As we observed that PhenDC3 induced an uncoupling between cellulo RG4s and mitochondria (Fig. 1a, b, Supplementary Fig. 1c), we asked whether the ligand-induced stabilization of the AKAP1 RG4s modified the translation-dependent localization of this mRNA to mitochondria. To this end, we used RNAscope[50] for the detection of AKAP1 mRNA and the MTND5 mitochondrial mRNA combined with immunofluorescence visualization of AKAP1 protein. As shown in Fig. 2d, e, puromycin induced a 20% loss of colocalization between the AKAP1 mRNA and the encoded protein comparable to that observed previously[15], with similar effects induced by PhenDC3 treatment. In agreement with previous findings[7] and with the results in Fig. 1a, b and Supplementary Fig. 1c that show ligand-induced uncoupling between the RG4 signal and mitochondria marker, we found that the association of AKAP1 mRNA with mitochondria was reduced in the presence of PhenDC3 to the same extent as in the presence of the translation inhibitor pur-omycin (Fig. 2f, g). Consistent with this, analysis of the distribution of AKAP1 mRNA colocalizing with the encoded protein between the mitochondria (MTND5 mRNA) and the cytosol revealed a predominant association with the mitochondria and a decrease in this fraction after treatment with the ligand, accompanied by a proportional increase in AKAP1 RNA/protein colocalization in the cytosol (Supplementary Fig. 3g). Similar losses of colocalization were observed for two other nuclear-encoded mitochondrial transcripts (Supplementary Fig. 3h) that contain RG4s (Fig. 2a). To strengthen the conclusion that PhenDC3 inhibits OMM-localized AKAP1 mRNA translation, we com-bined immunofluorescence visualization of TOMM20 with proximity ligation assay (PLA) detection of neo-synthesized AKAP1. We used antibodies against N-term AKAP1 and puromycin which is covalently incorporated into nascent protein chains. Consistent with Fig. 2d–g, we found that the PLA signal for the neo-synthetized AKAP1 was localized at the OMM, as opposed to the non-mitochondrial protein IkB (Fig. 2h, i and Supplementary Fig. 3i). PhenDC3 reduced both the number of PLA dots for the nascent AKAP1 (Supplementary Fig. 3j) and their OMM localization (Fig. 2i). Overall, these results indicate that RG4 stabilization decreased AKAP1 OMM-localized neo-synthesis.

To provide more direct evidence of RG4 involvement in OMM-localized translation, we generated mitochondria localization-specific

mRNA reporters containing the 3′UTR of GALNT2, a nuclear-encoded mitochondria mRNA in which (experimentally identified[18,19]) RG4s were either wild-type or mutated (Supplementary Fig. 3k). We observed that RG4 mutations reduced the luciferase expression in both Hela and HCT116 cells (Fig. 2j). Similar results were obtained with a reporter that did not form RG4s due to the guanines being replaced by the analogue 7-deaza-guanines (Fig. 2j), which prevents Hoogsteen base pairing and RG4 formation[51]. In agreement with this, colocaliza-tion between TOMM20 and PLA detection of neo-synthesized lucifer-ase reporters revealed a reduction in mitochondrially localized translation when the RG4 was unable to fold, being either mutated or containing the guanine analogue (Fig. 2k and Supplementary Fig. 3l). In line with this, RG4 formation decreased the colocalization between the GALNT2 mRNA reporter and the MTND5 mitochondrial mRNA (as revealed by RNAscope analysis) (Supplementary Fig. 3m). Taken together, these results suggest that RG4s are important determinants of OMM-localized translation and that the impairment of their dynamics between the folded and unfolded states interferes with this function, with consequences for mitochondrial respiration. Similarly to PhenDC3, treatment with cPDS induced RG4 stabilization (Supple-mentary Fig. 4a) and resulted in both reduced mitochondrial respira-tion (Supplementary Fig. 4b, c) and association of GALNT2 with the mitochondria (Supplementary Fig. 4d), indicating that the role of RG4 dynamics in the crosstalk between protein synthesis and energy metabolism is ligand-independent.

## hnRNP U binds to RG4-containing nuclear-encoded mitochondrial mRNAs

To gain insights into the proteins that could bind these RG4s, we reasoned that this factor must be both an RNA- and RG4-binding protein associated with the translation machinery. We thus intersected the list of RBPs interacting with the 3′UTRs of RG4-containing ribo-some-dependent nuclear-encoded mitochondrial mRNAs derived by ENCODE eCLIP (enhanced crosslinking immunoprecipitation)-data sets[52], the catalog of RG4-binding proteins[39] and the ribosome-associated proteins[53]. This analysis revealed that the hnRNP U, RPS3, and YBX3 RBPs could be strong candidates for binding to RG4s of nuclear-encoded mitochondrial mRNAs in a ribosome-dependent manner (Supplementary Fig. 5a). To further explore the regulatory mechanism of RG4s at OMM, we thus focused on hnRNP U as it was previously shown to play a role in energy metabolism[54,55].

hnRNP U is a protein of the hnRNP family, mainly described as a regulator of nuclear post-transcriptional steps of mRNA maturation[56–59]. However, its function related to RG4 regulation and mRNA translation has not been investigated yet. In agreement with previous data reporting the function of hnRNP U in the cytoplasm[60–64], we observed that, beyond its presence in nuclear fractions, hnRNP U was enriched in cytosolic and microsomal fractions (Fig. 3a). To investigate its binding to cytoplasmic RG4-containing mRNAs, we first reanalyzed previously published in cellulo hnRNP U-RNA interactions using crosslinking and immunoprecipitation (CLIP) of hnRNP U from

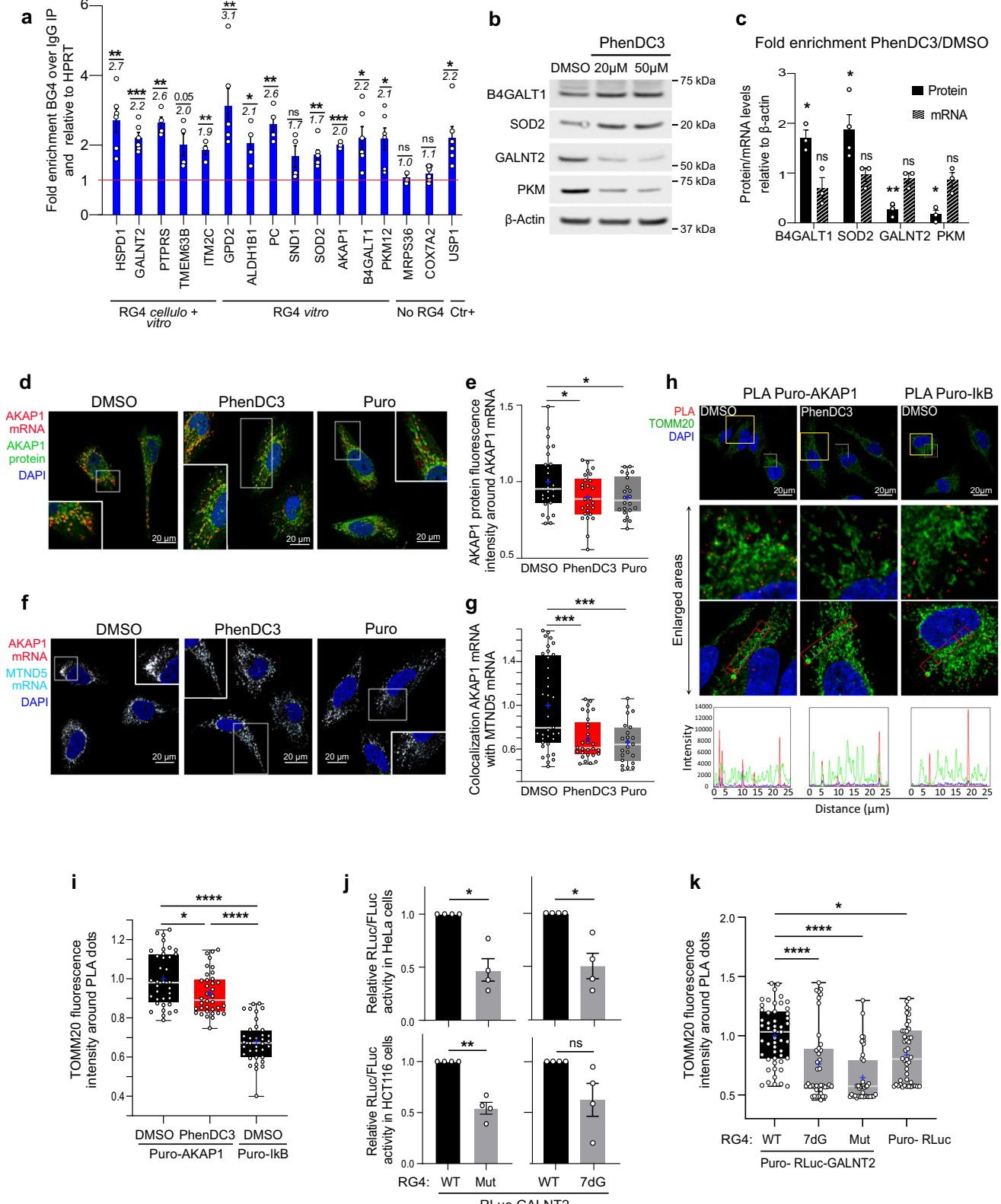

cytoplasmic fractions followed by deep RNA sequencing (cyto-CLIP[62]). Strikingly, 11.7% of 793 hnRNP U-bound mRNAs in the cytoplasm were associated with mitochondria (Fig. 3b) and significantly associated with metabolic reprogramming in cancer by pathway enrichment analysis (Fig. 3c). Consistently, hnRNP U immunoprecipitation with cytoplasmic extracts followed by RT-qPCR analysis revealed that hnRNP U associated with nuclear-encoded mitochondrial mRNAs in

cellulo (Fig. 3d, Supplementary Fig. 5b). To determine if hnRNP U binds to RG4s, we first used RNA affinity chromatography with cytoplasmic extracts. Consistent with a recent proteomic screen for cytoplasmic proteins binding to folded/unfolded RG4s[26], we found that hnRNP U preferentially bound to canonical (G3A2G3A2G3A2G3) or naturally occurring sequences (i.e., the RG4-forming sequence in the 5'UTR of NRAS[25]) over RNAs that do not form RG4s due to the replacement of

**Fig. 2 | RG4s are molecular determinants of OMM-localized translation.**
**a** Immunoprecipitation (IP) of in cellulo RNA-protein complexes using the BG4 antibody or control IgG, followed by RT–qPCR analysis. Fold changes are indicated. Data represent n = 2 or n = 3 independent experiments in duplicate when 4 or 6 dots are plotted, respectively. **b** Western blot analysis in HCT116 cells treated with PhenDC3 for 48 h. **c** Fold enrichment of protein/mRNA levels from (**b**). **d** AKAP1 mRNA/protein localization analysed by RNAscope and immunofluorescence (IF). (**e**) Quantification from (**d**). *P*-value = 0.0297 or 0.0425 for PhenDC3 or Puro treatment, respectively (two-sided paired t-test). **f** RNAscope analysis of AKAP1 and MTND5 mRNA localization as in (**d**). **g** Quantification from (**f**). **h** Proximity Ligation Assay coupled with puromycin Labeling (PLA-Puro) of AKAP1 or IkB, combined with TOMM20 IF. Graphs displaying the fluorescence intensity in each channel over the distance are depicted. **i** Quantification from (**h**). *P*-value = 0.0125 for puro-AKAP1 after PhenDC3 treatment. **j** Ratio of Renilla/Firefly luciferase activities (Rluc/Fluc)

after transfection of RLuc-GALNT2 mRNA reporters (depicted in Supplementary Fig. 3j) containing the RG4 unmodified (WT), mutated (Mut) or 7-deaza-modified (7dG) for n = 4 independent experiments. **k** Quantification of the colocalization between TOMM20 and PLA-Puro dots from in vitro transcribed Rluc-GALNT2 reporter mRNA containing the RG4 WT, Mut or 7dG from images in Supplementary Fig. 2i. *P*-value = 0.0243 for the Puro-Rluc. In (**a**, **c**, **j**), data represent mean ± SEM. In (**e**, **g**, **i**, **k**), boxes extend from 25th to 75th percentiles; middle line is the median, and "+" is the mean; the whiskers show upper and lower extremes. For (**a**, **b**, **c**, **e**, **g**, **i**, **j**, **k**) Source Data are provided as a Source Data file indicating exact *P*-values for (**a**, **c**, **j**). In (**b**, **c**), the blot shown is representative of n = 3 independent experiments. In (**d**, **f**, **h**) a representative field from n = 3 independent experiments is shown. In (**e**, **g**, **i**, **k**), data represent n = 3 independent experiments and each dot is a cell. In (**a**, **c**, **g**, **j**, **k**), *P < 0.05, **P < 0.01, ***P < 0.001, ****P < 0.0001, ns non-significant (two-sided paired t-test).

guanines with 7-deaza-guanines (Fig. 3e and Supplementary Fig. 5c). Furthermore, previous RNA affinity purification followed by mass spectrometry (RP-MS) experiments identified hnRNP U as being preferentially associated with wild-type as compared to sequences lacking G-runs[27–29,31] or forming hairpin structures[24]. These results thus support the conclusion that hnRNP U is an RG4-binding protein preferentially binding to folded RG4s. To explore these results further, we intersected the hnRNP U cyto-CLIP dataset with both in vitro validated RG4s (datasets from Guo et al.[18] and Kwok et al.[19]) and in silico predicted RG4s[39]. As shown in Fig. 3f, the hnRNP U-bound 3'UTR and CDS regions contain a similar proportion of experimental (based on Kwok et al.[19]) and predicted RG4s, while the proportion in the 5'UTR was consistently smaller compared to CDS/3'UTR. We found that 44.64% and 34.68% of hnRNP U-bound mRNAs contained experimentally validated RG4s from the Kwok[19] and Guo datasets[18], respectively, as extracted from the QUADRatlas database[39]. This proportion increases to 61% when considering only the high-scoring RG4 forming sequences predicted with QGRSmapper[65] (score ≥ 19). Importantly, hnRNP U-bound mRNAs containing RG4s identified experimentally or in silico predicted were significantly associated with metabolic reprogramming pathways (adjusted *P*-value 3.2E-08 (Kwok dataset), 2.5E-07 (Guo dataset), 9.4E-06 (predicted)). In addition, analysis of the density of hnRNP U binding sites around RG4s (based on Kwok et al.[19]) indicated significant colocalization for the CDSs and the 3'UTRs but not for the 5'UTRs (Fig. 3g). To define whether this pattern is specific to hnRNP U, we looked at the density of binding around RG4s for other ENCODE eCLIP-profiled RBPs. We found that 62/132 RBPs (47%) do not have the same binding pattern as hnRNP U (as exemplified by the RBP NCBP2 (Supplementary Fig. 5d)). Finally, RIP assays with cytoplasmic extracts and the BG4 antibody confirmed that several nuclear-encoded mitochondrial mRNAs targeted by hnRNP U can form RG4s in cellulo (Fig. 3h). Based on the "bind, unfold and lock" model[26], whereby RG4-BPs synergize with helicases ("bind") to unwind RG4s ("unfold") and promote G-rich-BP binding that keeps RG4s unfolded ("lock"), the hnRNP U-RG4s interaction could cooperate with RNA helicases, thus inducing RG4 melting. In support of this hypothesis, we found that RG4 foci visualized using the BioCyTASQ increase after hnRNP U silencing (Fig. 3i, j). This is consistent with what we observed after treatment with PhenDC3 (Fig. 1a, b, Supplementary Fig. 1c, g). Overall, these results suggest that cytoplasmic hnRNP U binds nuclear-encoded RG4-containing mitochondrial mRNAs with consequences on the RG4 structural equilibrium.

### hnRNP U localizes to the OMM and interacts with the translational machinery and nuclear-encoded mitochondrial proteins

To further explore the role of hnRNP U when bound to RG4-containing nuclear-encoded mitochondrial mRNAs, we reasoned that it should be localized in OMM compartments and possibly interact with the factors therein contained, i.e. nuclear-encoded mitochondrial mRNAs to be translated, ribosomes, translational

regulators, and mitochondrial proteins to be imported into mitochondria[66,67]. To address these possibilities, we purified mitochondria and determined whether hnRNP U was associated with these organelles. We observed that hnRNP U was found in the cytoplasm, microsomes, and mitochondrial fractions (Fig. 4a) like other proteins associated with OMM (TOMM20), residing in the mitochondrial matrix (DHX30, GRSF1), or associated with the inner mitochondrial membrane (OXPHOS), but unlike other RG4-BPs (DDX5) or known hnRNP U interactors (PRMT1[68]). Then, we asked whether hnRNP U could be associated with the OMM using the protease protection assay after mitochondria purification. In the absence of a detergent that solubilizes the mitochondrial membrane (Triton-X), the addition of proteinase K removed hnRNP U from the mitochondrial fraction, suggesting that hnRNP U is an OMM-interacting protein (Fig. 4b). DDX3X, a RG4 helicase known to play a role in mRNA translation and mitochondrial respiration[69], showed a proteinase K protection profile similar to that of hnRNP U (Fig. 4b). Unlike hnRNP U and DDX3X, a fraction of cytoplasmic GRSF1 and DHX30, two RBPs involved in mRNA metabolism within mitochondria[38,70,71], was partially resistant to proteases but completely degraded when mitochondria were permeabilized with Triton-X (Fig. 4b). These results suggest that hnRNP U and DDX3X do not reside in mitochondria but rather are associated with the OMM where they can interact with RG4-containing mRNAs. This is in agreement with a proteomic analysis of purified mitochondria from HCT116[67]. Indeed, consistent with Fig. 3e and our previous proteomic screen for proteins binding folded/unfolded RG4s[26], RNA pull-down performed with mitochondrial (Supplementary Fig. 6a) or OMM fractions (Fig. 4c) showed that hnRNP U and DDX3X interact preferentially with folded RG4s in these fractions. In contrast, GRSF1, which promotes RG4 melting[38] and binds to G-rich sequences[72], showed a stronger association with G-rich sequences unable to form RG4s (Fig. 4c, Supplementary Fig. 6a).

Then, we analyzed the hnRNP U protein interactome by performing immunoprecipitation of hnRNP U partners from cytoplasmic fractions in the presence of benzonase, an endonuclease that removes nucleic acids. Proteins co-immunoprecipitated with hnRNP U were subjected to tryptic digestion followed by higher-energy collisional dissociation-tandem mass spectrometry (HCD-MS/MS), thus allowing quantitative label-free proteomic analysis of protein–protein interaction data[73]. The results were evaluated using significance analysis of interactome (SAINT) analysis[74], a computational tool that assigns confidence scores to protein–protein interactions, with hnRNP U interactors defined as proteins with SAINT score SP ≥ 0.8. This quantitative analysis with four replicates enabled the identification of 134 proteins that interact with hnRNP U in an RNA-independent manner (Fig. 4d, Supplementary Data 1), assigned to specific functional pathways such as translation and mitochondrial translation (Supplementary Fig. 6b). Among these proteins (Fig. 4d), we found 1) ribosomal proteins of the cytoplasmic and mitochondrial translational

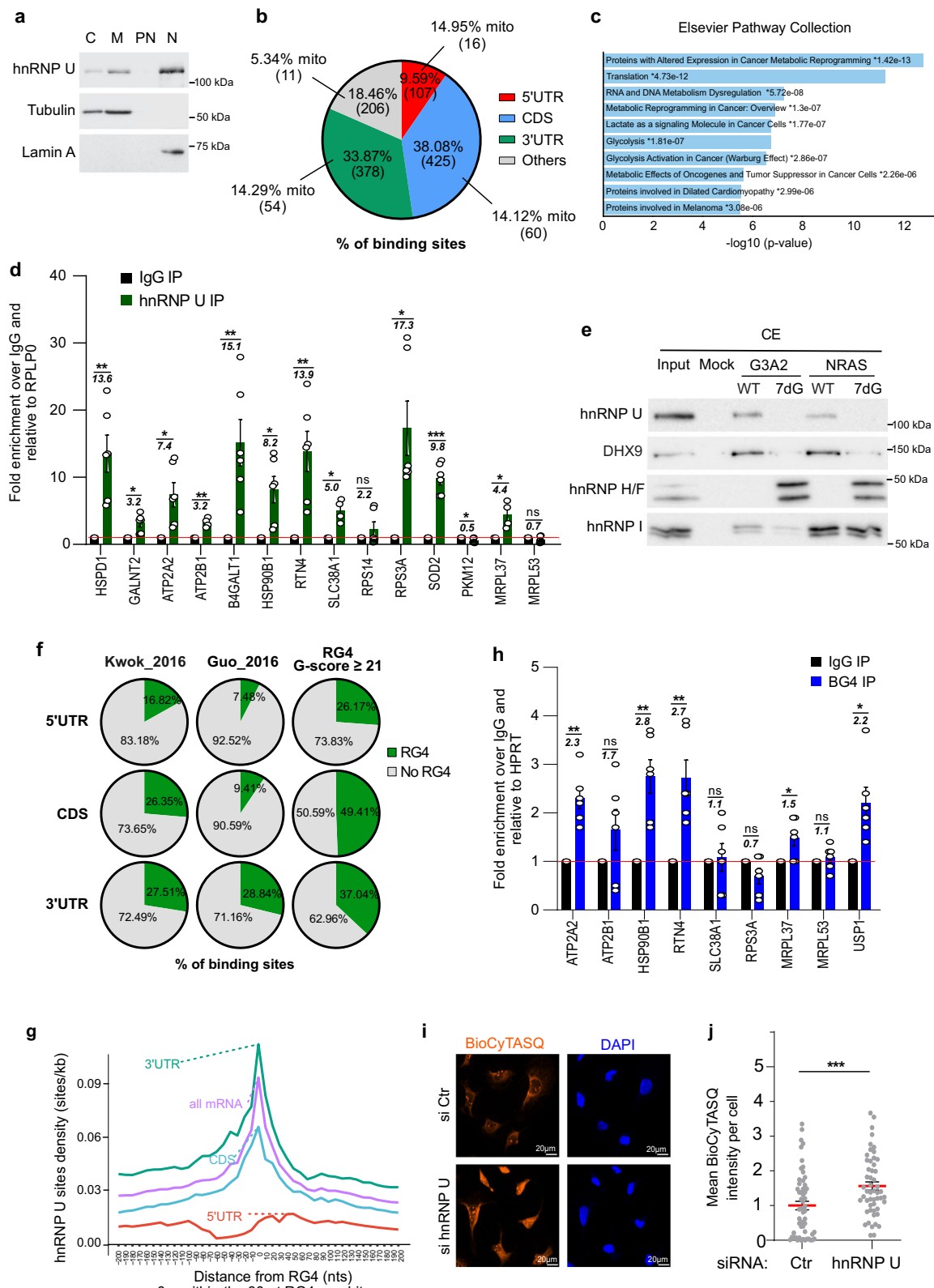

machineries, some of which were validated by immunodetection (Supplementary Fig. 6c), and 2) 26 RG4-binding proteins, including DDX3X, which, similarly to hnRNP U, was associated with the OMM (Fig. 4b) and bound RG4s at this specific subcellular location (Fig. 4c, Supplementary Fig. 6a). Taken together, these results suggest that a fraction of hnRNP U is localized at the OMM where it interacts with both the translational machinery (including ribosomes, RG4 binding

proteins, and helicases as DDX3X) and nuclear-encoded mitochondrial proteins.

## hnRNP U regulates the OMM-localized protein synthesis of RG4-containing mRNA targets

The observation that hnRNP U was found in the cytoplasm (Fig. 3a) where it interacts with ribosomal proteins and translational regulators

**Fig. 3 | hnRNP U binds nuclear-encoded RG4-containing nuclear-encoded mitochondrial mRNAs. a** Subcellular fractionation of HCT116 cells, followed by western blot analysis. Lamin A: nuclear marker, Tubulin: cytoplasmic marker associated with microsomes. Nuclear (N), perinuclear (PN), microsomal (M), and cytosolic fractions (C). Shown is a representative result from n = 3 independent experiments. **b** Proportion of hnRNP U binding sites in 5′UTR, CDS, and 3′UTR and percentage of mitochondrial mRNAs (mito) bound by hnRNP U in each region. **c** −log10(P-value) of the enrichment analysis of hnRNP U targets[61]. P-value computed from Fisher exact test. **d** Immunoprecipitation of cellulo RNA-protein complexes (RIP) using the hnRNP U antibody, followed by RT-qPCR analysis. Fold changes are indicated in italics. **e** RNA affinity chromatography using the G3A2 and NRAS RG4s native (WT) or 7-deaza-modified (7dG), followed by western blot analysis. Shown is a representative result from n = 6 and n = 2 independent experiments for the G3A2 and NRAS RG4s, respectively. **f** Proportion of hnRNP U binding sites in 5′UTR, CDS, and 3′UTR containing experimental and predicted RG4s. **g** Density of hnRNP U binding sites around RG4s located in mRNAs (all mRNA), 5′UTRs, CDSs, and 3′UTRs. At distance 0 and until +30 −80: P = 0.004 for CDS, P = 0 for 3′UTR, P = 1 for 5′UTR (Bootstrap test with 1000 sets of random sites). **h** RIP using the BG4 antibody, followed by RT-qPCR analysis. Fold changes are indicated in italic. USP1: positive control[26]. **i** RG4 detection after control (Ctr) or hnRNP U siRNAs. Scale bar: 20 μm. Shown is a representative result from n = 4 independent experiments. **j** Quantification of signal intensities from (**i**), each dot is a cell, P-value = 0.000181 (two-sided paired t-test). For (**a, d, e, h, j**) source data are provided as a Source Data file indicating exact P-values for (**d**) and (**h**). When indicated, *P < 0.05, **P < 0.01, ***P < 0.001, ns: non-significant (two-sided paired t-test). In (**d, h**), data are presented as mean values ± SEM of n = 2 or n = 3 independent experiments in duplicate when 4 dots or 6 dots are plotted, respectively.

(Fig. 4d) prompted us to study its role in translation. To this end, we performed polysome profiling combined with immunoblotting to monitor the distribution of hnRNP U in translationally inactive and active fractions in the presence or absence of puromycin, a drug-inducing ribosome dissociation. We found that hnRNP U co-sedimented with translating polyribosomes and that their association depended on polysome integrity (Fig. 5a). To demonstrate a functional role for hnRNP U in the regulation of translation, we quantified the global protein synthesis after hnRNP U siRNA-mediated silencing by pulsed puromycin labeling and immunoblotting with an anti-puromycin antibody (i.e. SUnSET). We found that the depletion of hnRNP U did not modify the overall translation rates (Supplementary Fig. 7a, b) while the polysomal profile was only slightly altered by hnRNP U depletion (Supplementary Fig. 7c, d), indicating that hnRNP U-deficient cells are not globally deficient in protein synthesis. Since in these conditions cellular proliferation was not affected (Supplementary Fig. 7e), changes in translational efficiency did not depend on this process. Based on these results and our previous findings showing hnRNP U binding to RG4-containing nuclear-encoded mRNAs (Fig. 3d, f, g, h), we tested whether depletion of hnRNP U could selectively control the translation of these mRNAs. To this end, we performed polysomal fractionation of hnRNP U-depleted cells, followed by RNA isolation from non-polysome (NP), light (LP) and heavy (HP) polysome fractions, and then by either RNA sequencing (Supplementary Fig. 7f) and/or RT-qPCR analysis (Fig. 5b, Supplementary Fig. 7g, h). Following hnRNP U silencing, the translational efficiency of these mRNAs was quantified either by analyzing the HP/total mRNA ratio (Fig. 5b, Supplementary Fig. 7g) or by measuring the distribution of each mRNA across the gradient (Supplementary Fig. 7h). Focusing first on previously characterized RG4-containing nuclear-encoded mitochondrial mRNAs bound by hnRNP U (Fig. 3d), we observed that hnRNP U depletion induced a significant decrease in mRNA association with translating polysomes (Fig. 5b, Supplementary Fig. 7g, h), indicating a role of hnRNP U in translational activation of a subset of nuclear-encoded mitochondrial mRNAs. To strengthen these results and provide evidence of direct involvement of hnRNP U in mRNA translation, we performed in vitro translation using purified recombinant hnRNP U and a luciferase reporter transcript containing the 3′UTR of GALNT2 (Supplementary Fig. 3g), an RG4-containing transcript (Fig. 2a) bound (Fig. 3d) and translationally regulated by hnRNP U (Fig. 5b, Supplementary Fig. 7h). We found that hnRNP U addition increased the translation of the mRNA reporter containing the wild-type RG4 but not the mutated version, nor the luciferase control transcript (Fig. 5c). Furthermore, similar to PhenDC3 treatment (Fig. 2d, e), hnRNP U silencing reduced the colocalization between AKAP1 mRNA, which is a translational target of hnRNP U (Fig. 5b, Supplementary Fig. 7h), and the encoded protein (Fig. 5d–g), suggesting that hnRNP U regulates OMM-localized mRNA translation. To study the extent of translational regulation of nuclear-encoded mitochondrial mRNAs by hnRNP U, we analyzed the translational targets of hnRNP U at the transcriptome-wide level using polysomal profiling followed by RNA sequencing. We identified 60 RG4-containing nuclear-encoded mitochondrial transcripts whose association with HP fractions was altered after hnRNP U depletion, a subset of which was validated by RT-qPCR (Supplementary Fig. 7f, g, h). Next, we defined whether the hnRNP U-dependent mechanism of mitochondrial mRNA translational regulation could be tuned in response to stimuli inducing mitochondrial metabolic switches. A strong candidate for this function is the mechanistic/mammalian serine/threonine kinase target of rapamycin (mTOR) as its inhibition selectively inhibited the translation of a subset of nuclear-encoded mitochondrial mRNAs[75] and reduced the association of hnRNP U with mRNAs[76]. In agreement with previous results on stress-induced RG4 folding in cancer cells[77], we observed that rapamycin, an mTOR inhibitor mimicking the effect of starvation, induced a significant increase in RG4 structuration (Supplementary Fig.8a). Importantly, rapamycin or Torin 1, a very potent and selective mTOR inhibitor, reduced the colocalization between AKAP1 mRNA and the encoded protein (Fig. 5f, g and Supplementary Fig. 8b). Finally, we found that hnRNP U binding to RG4s (Fig. 5h, i) and the translation machinery (Supplementary Fig. 8c) was inhibited by Torin 1, possibly through altered phosphorylation of hnRNP U (Supplementary Fig. 8d, e). Together, these results suggest that hnRNP U is associated with the translational machinery and regulates the OMM-localized protein synthesis of RG4-containing mRNA targets with functions in mitochondria. Importantly, our data suggest the existence of a mTOR-hnRNP U-mitochondrial translation axis that modulates cellular energy production by mitochondria. The underlying mechanism involves mTOR stimulating the selective translation of nuclear-encoded mitochondrial mRNAs by modulating RG4 formation as well as hnRNP U phosphorylation and polysomal association and its interaction with RG4-containing mRNAs.

## hnRNP U affects oxidative metabolism by regulating the synthesis of OXPHOS complex proteins

OXPHOS, the main energy source of the cell, involves five multisubunit protein complexes harboring more than 80 nuclear-encoded proteins, as well as 13 mitochondrial-encoded subunits (complexes I, III, IV, and V). According to a recently proposed model of mitochondrial translational plasticity[78], the synthesis of the latter adapts to the influx of nuclear DNA-encoded OXPHOS subunits. Based on this model and taking into account that hnRNP U interacts with mRNAs and proteins (Fig. 3b, d) whose products are imported into the mitochondria, we asked whether hnRNP U shaped OXPHOS assemblies and affected mitochondrial functions. We observed that hnRNP U silencing reduced the expression of proteins belonging to the five OXPHOS complexes, including both nuclear-encoded (ATP5A, UQCRC2, SDHB, NDUFB8) and mitochondria-encoded proteins (COX2) (Fig. 6a, b). We then determined whether hnRNP U-induced altered expression of OXPHOS complexes resulted in modified OXPHOS enzymatic activity measured by OCR rates using Seahorse assays. Figure 6c, d shows that hnRNP U

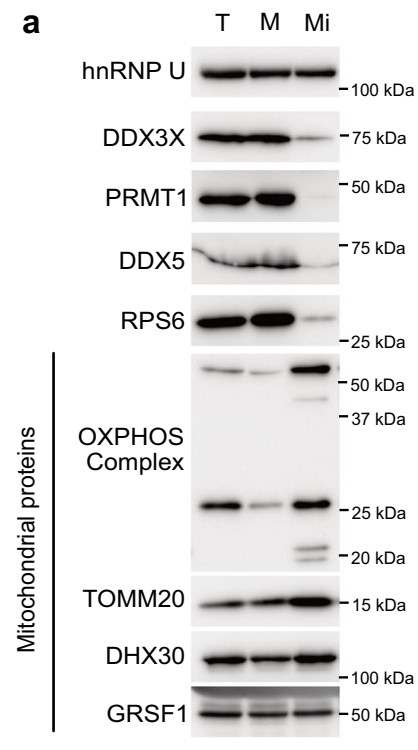

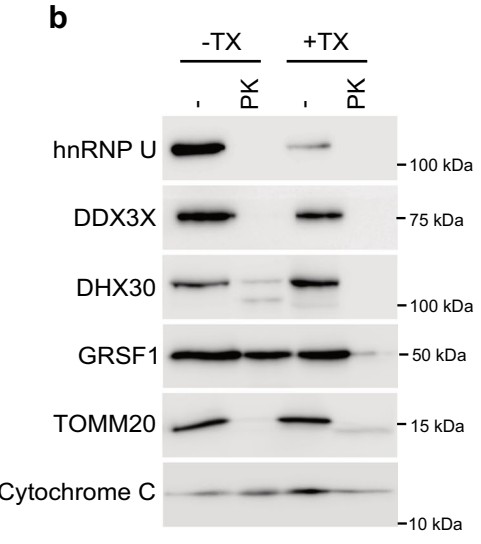

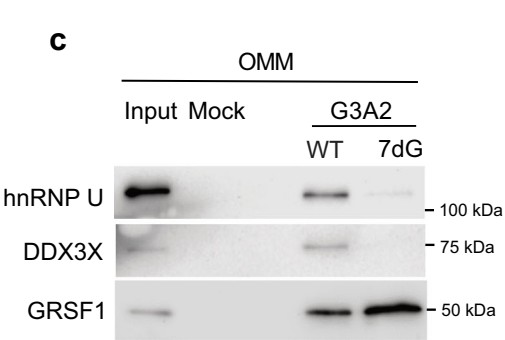

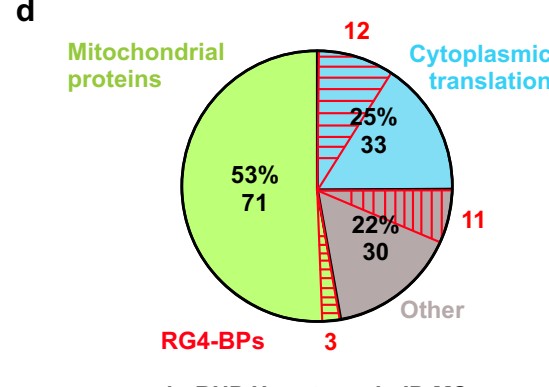

**hnRNP U partners in IP-MS**

**Fig. 4 | hnRNP U protein interactors and sub-cellular localization. a** Subcellular fractionation of HCT116 cells to obtain Total (T), Microsomal (M), and Mitochondrial (Mi) fractions, followed by western blot analysis. Shown is a representative result from n = 8 independent experiments for hnRNP U and DDX3X, n = 5 independent experiments for RPS6, OXPHOS complex and GRSF1, n = 6 independent experiments for TOMM20 and DHX30, n = 2 independent experiments for PRMT1 and DDX5. **b** Protease protection assay after mitochondria purification of HCT116 cells followed by western blot analysis of the indicated proteins. TX: Triton-X. PK: Proteinase K. Shown is a representative result from n = 2 independent experiments. **c** RNA affinity chromatography using the G3A2 sequences either native (WT) or

7-deaza-modified (7dG) and HCT116 OMM (Outer Mitochondrial Membrane) extracts, followed by western blot analysis. Shown is a representative result from n = 4 independent experiments for hnRNP U and GRSF1 and n = 3 independent experiments for DDX3X. **d** Distribution of hnRNP U partners identified by immunoprecipitation (IP) cytoplasmic HCT116 cell extracts, followed by mass spectrometry (IP-MS, n = 4 independent experiments). $P = 1.58e\text{-}46$, $P = 5.01e\text{-}46$, and $P = 1.58e\text{-}22$ for the mitochondrial proteins, the proteins involved in the cytoplasmic translation, and the RG4-BPs, respectively (Fisher test). For (**a**, **b**, **c**), source data are provided as a Source Data file.

depletion significantly impairs mitochondrial respiration with reduced basal OCR and lowered ATP turnover rate. We also tested the effect of hnRNP U depletion on glycolysis because (1) the inhibition of OXPHOS function might induce a metabolic shift to glycolysis-mediated energy production, also called Warburg effect[79], and (2) hnRNP U-bound mRNAs are significantly associated with glycolysis-related pathways (Fig. 3c). Using the Seahorse mitochondrial and glycolytic stress tests as a measure of glycolysis, we found that silencing of hnRNP U inhibited glycolysis (Fig. 6e, Supplementary Fig. 9a). Since DDX3X, similarly to hnRNP U, was found at the OMM (Fig. 4a, b), interacts with RG4s (Fig. 4c, Supplementary Fig. 6a) and was shown to modulate

mitochondrial respiration[69], we tested the effect of silencing DDX3X alone or in combination with hnRNP U on both mitochondrial respiration and glycolysis. We observed that DDX3X and hnRNP U similarly reduced mitochondrial respiration and glycolysis, and that combined depletion led to a further reduction, though not in a synergistic manner (Fig. 6c, d, Supplementary Fig. 9a). We also found that the effect of hnRNP U silencing on mitochondrial respiration and glycolysis can be rescued by re-expression of an RNAi resistant version of hnRNP U at a level proportional to the amount of ectopic hnRNP U, ruling out the possibility of an off-target effect in the analysis (Supplementary Fig. 9b–f). Consistently, hnRNP U depletion also reduced

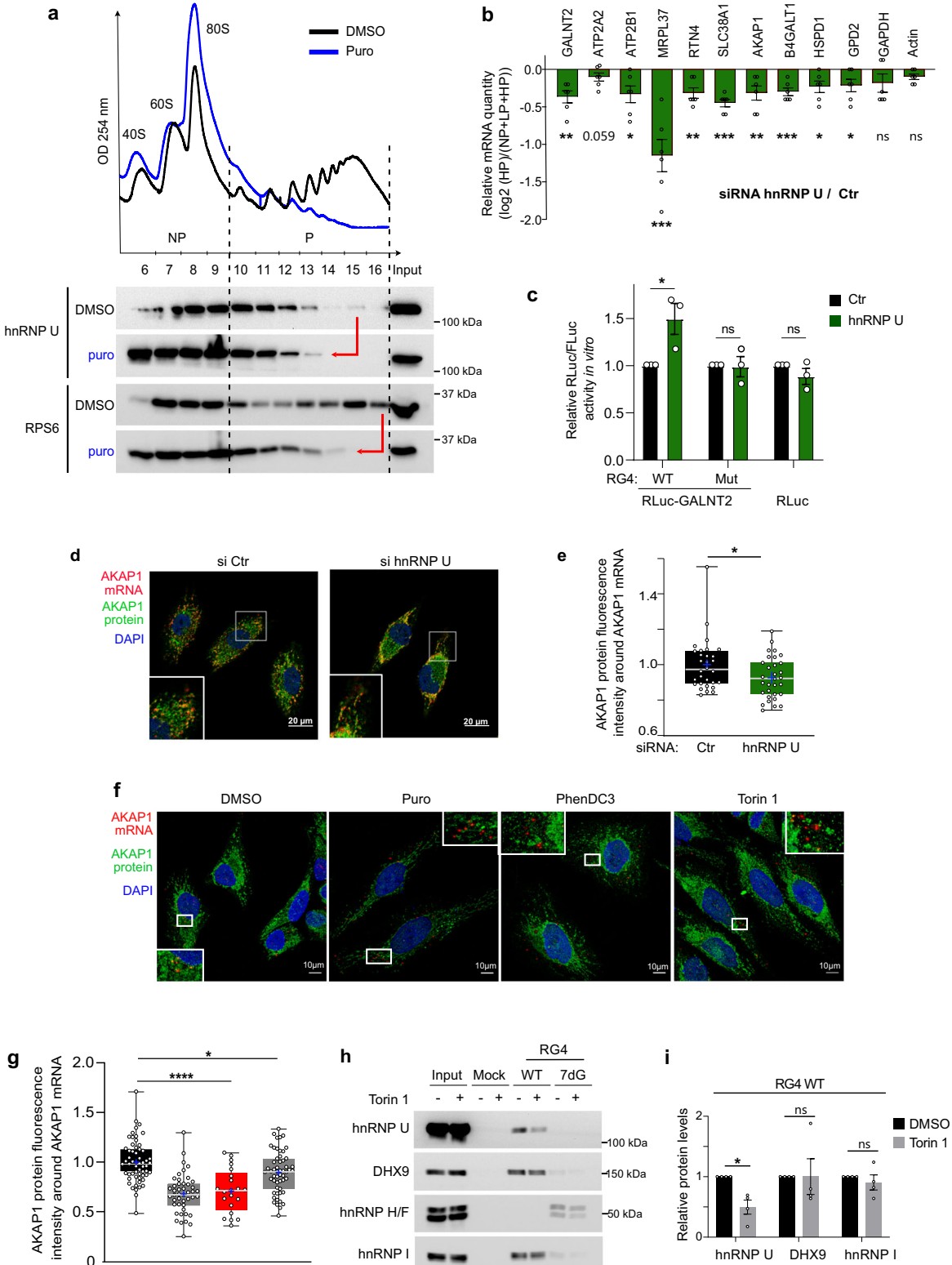

cell proliferation (Fig. 6f). Since depleting hnRNP U increased RG4 stability (Fig. 3i, j), we wondered whether this effect was also observed after ligand-mediated stabilization. Indeed, Supplementary Fig. 9g shows that PhenDC3 reduced glycolysis in three different cell lines, suggesting that RG4 stabilization induced by either small molecule ligands or hnRNP U-silencing may have a dual metabolic effect by inhibiting oxidative metabolism, both glycolysis and OXPHOS.

## hnRNP U, DDX3X and GRSF1 cooperate in the translational regulation of RG4-containing nuclear-encoded mitochondrial mRNAs

We then sought to define the molecular mechanism underlying the function of hnRNP U in translation regulation of RG4-containing nuclear-encoded mitochondrial mRNAs. The observation that hnRNP U and DDX3X, 1) interact (Supplementary Data 1), in agreement with

**Fig. 5 | hnRNP U is involved in translation regulation of RG4-containing nuclear-encoded mitochondrial mRNAs. a** Polysome profile of HCT116 cells DMSO or puromycin (Puro) treated, followed by western blot analysis. RPS6: positive control. Shown is a representative result from n = 2 independent experiments. **b** RT-qPCR analysis from non-polysomal (NP), light (LP) and heavy (HP) polysomal fractions from Supplementary Fig. 7c. *P < 0.05, **P < 0.01, ***P < 0.001, ns: non-significant (two-sided paired t-test). **c** In vitro translation of Rluc-GALNT2 reporter mRNAs (depicted in Supplementary Fig. 3j) containing the RG4 unmodified (WT), or mutated (Mut) and of the control Rluc reporter mRNA with or without recombinant hnRNP U. **d** RNAscope analysis of AKAP1 mRNA localization and immunofluorescence (IF) analysis of AKAP1 protein localization in HeLa cells treated with control (Ctr) or hnRNP U siRNAs. Scale bar: 20 μm. **e** Quantification from images in (**d**). Data represent n = 3 independent experiments and each dot is a cell, P-value = 0.02486 (two-sided paired t-test). (**f**) RNAscope and IF analysis as in (**d**) in HeLa cells treated 20 μM PhenDC3 for 24 h or 300 nM Torin 1 for 2 h. Scale bar: 10 μm. **g** Quantification from images in (**f**). Data represent n = 3 and n = 2 independent experiments for the Puro, Torin 1 and PhenDC3 treatments, respectively, each dot is a cell. **h** RNA affinity chromatography using RG4 sequences either native (WT) or 7-deaza-modified (7dG) and cytoplasmic extracts of HCT116 treated with DMSO or 300 nM Torin 1 for 3 h, followed by western blot analysis. **i** Quantification of the protein levels on the RG4 WT from (**h**). P-value = 0.02336, ns non-significant (two-sided paired t-test). For (**a**, **b**, **c**, **e**, **g**, **h**), Source Data is provided as a Source Data file indicating exact P-values for (**b**, **c**, **g**). Data are presented as mean values ± SEM of n = 3 for (**b**), (**c**), n = 4 for (**i**) independent experiments. For (**e**, **g**), the box extends from 25th to 75th percentiles; middle line is the median and "+" is the mean; the whiskers show upper and lower extremes. For (**d**, **f**), shown is a single representative field over n = 3 independent experiments.

Brannan et al.[80]), 2) are both recruited on folded RG4s in OMM fractions (Figs. 3, 4c) have a similar function in mitochondrial respiration and glycolysis (Fig. 6c, d, Supplementary Fig. 9a) prompted us to investigate whether hnRNP U functions cooperatively with DDX3X to control mitochondrial mRNA translation and function. We first studied the molecular determinants enabling hnRNP U to interact with the RNA and DDX3X. Consistent with the evidence that the RGG (arginine-glycine-glycine) domain is involved in protein-RG4[81] and protein-protein interactions[82], we found that deletion of this domain in the C-terminal region of hnRNP U decreased its interaction with both DDX3X (Fig. 7a) and RG4s (Fig. 7b), whereas DDX3X binding to RG4s was unaffected (Fig. 7b), suggesting that DDX3X may bind to RG4s independently of hnRNP U. If the "bind, unfold, lock" model involving DHX36 and hnRNP H/F that we proposed previously[26] also applies here, hnRNP U and DDX3X could be recruited to RG4s first, followed by GRSF1, a member of the hnRNP H/F family, which binds to unfolded RG4s in the OMM fractions (Fig. 4c), to maintain them in the unfolded conformation. This would agree with data showing that GRSF1 (1) is found in OMM fractions (Fig. 4b) where it associates with G-rich sequences unable to form RG4s (Figs. 4c), (2) it is involved in binding and unwinding G4 structures[38,83], (3) predominantly binds to nuclear-encoded mRNAs[72], specifically to RG4s within 5' and 3'UTRs[84] and (4) regulates mRNA translation via 5' and 3'UTR mechanisms[85–87]. Furthermore, this scenario would be consistent with findings showing that DDX3X, 1) binds to RG4s in mRNAs encoding oxidative phosphorylation proteins[25], (2) plays a role in translation when associated with 3'UTRs[88], and (3) impacts on mitochondrial functions[69]. If the model is true, we would expect that the depletion of proteins that resolve RG4s and those that keep them unfolded would lead to an increase in RG4 formation. In agreement with this, we found that RG4 foci visualized using BioCyTASQ increased after DDX3X and GRSF1 silencing (Supplementary Fig. 10a,b). Then, we performed RIP assays using antibodies against GRSF1 or DDX3X and cytoplasmic extracts, which we combined with hnRNP U silencing to determine the cooperativity among these factors. We observed that DDX3X bound to the same targets as hnRNP U, and, in agreement with Fig. 7b, that depleting hnRNP U did not alter the binding of DDX3X to these targets (Fig. 7c, Supplementary Fig. 10c). This suggests that, as previously demonstrated for the hnRNP H/F RBP and the DHX36 helicase[26], DDX3X binds first followed by hnRNP U. This possibility was confirmed by RG4 pull-down analysis of hnRNP U from extracts depleted or not of DDX3X, showing that in the absence of DDX3X the binding of hnRNP U to RG4s was reduced but not that of another RG4-binding protein, DHX36 (Fig. 7d), suggesting a specific recruitment of hnRNP U by DDX3X bound to RG4s. GRSF1 was also unaffected by the loss of DDX3X binding (Fig. 7d), which is consistent with the fact that it is a G-rich-BP but also that it did not interact directly with hnRNP U, as shown by co-immunoprecipitation analyses (Fig. 7a, e) and proteomic analysis of hnRNP U interactants (Supplementary Data 1). These results, together with a trend for GRSF1, similar to DDX3X, to bind to the same hnRNP U targets (Supplementary Fig. 10d, e), suggest that binding of the hnRNP U-DDX3X complex and GRSF1 probably involves other factors to facilitate the dynamics of RG4 unfolding.

To determine whether the physical interaction between DDX3X and hnRNP U resulted in a functional interplay, we first investigated the possibility of a cross-regulation between the two proteins and then compared the effect of their co-depletion with that of individual hnRNP U or DDX3X silencing on mitochondrial protein expression. We found that inhibition of hnRNP U did not alter DDX3X expression and vice versa (Fig. 7f). Individual inhibition of either protein had a similar effect on the expression of two OXPHOS proteins (Fig. 7g, h) which is consistent with the findings that hnRNP U and DDX3X share a subset of mitochondrial mRNA targets. The observation that co-silencing of the two proteins did not have a significantly greater effect than depletion of hnRNP U and DDX3X individually (Fig. 7g, h) suggests a sequential recruitment model rather than a synergistic one in which binding of one would potentiate that of the other. This conclusion is also supported by mitochondrial and glycolysis assays after individual and co-depletion of hnRNP U and DDX3X (Fig. 6c, d, Supplementary Fig. 9a). Taken together, our results suggest that hnRNP U, DDX3X, and GRSF1 are players in RG4-dependent OMM localized translational regulation impacting mitochondrial function and energy metabolism.

## Discussion

Gaining insight into mitochondrial protein synthesis is essential to better untangle the link between mitochondrial metabolism and cancer development and to identify novel vulnerabilities that can be targeted for therapeutic purposes[2]. Emerging evidences support the notion of compartmentalized translation and co-translational targeting, which relies on translation factories providing a means of regulating the metabolism of nascent proteins where specific mRNA subsets are translated[15]. One key question to go further in this concept and explore its clinical applicability is what are the molecular determinants driving this specificity. A number of RBPs have been shown to play a role[47,89] but the *cis* determinants remain poorly characterized, with the role of RNA structures not yet proven. We propose here that RG4 structures are involved in mitochondrial mRNA translation and that their role in protein synthesis, association with OMMs, and mitochondrial function depend on their structural dynamics that are regulated by hnRNP U and modulated by small-molecule ligands. In our model (Fig.8), when the unfolded form is favored co-translational localization of nuclear-encoded mitochondrial mRNAs occurs at the OMM, allowing mitochondrial proteins to be synthesized and imported into the mitochondria, resulting in efficient mitochondrial respiration. However, shifting the structural equilibrium to a stably folded RG4 conformation, or failing to form an RG4, inhibits both translation and cytoplasmic protein expression of nuclear-encoded mitochondrial mRNAs at the OMM. This effect is associated with the physical uncoupling of the mitochondrial mRNA from both the encoded protein and mitochondria, resulting in the inhibition of

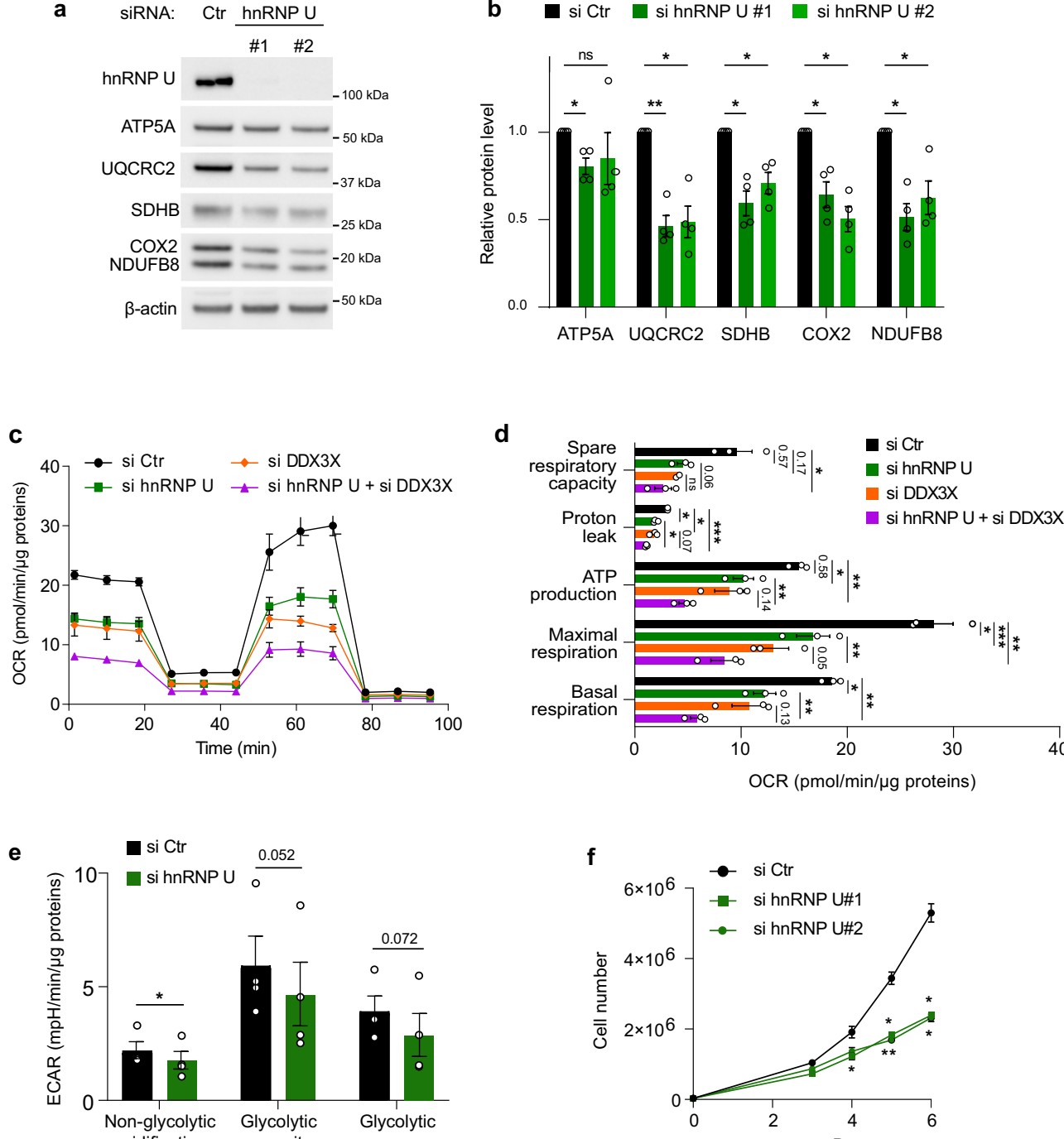

**Fig. 6 | hnRNP U modulates OXPHOS expression and function. a** Western blot analysis in HCT116 cells treated with control (Ctr), and 2 different hnRNP U (#1, #2) siRNAs for 72 h. The blot shown is a representative result from n > 4 independent experiments. **b** Quantification of the protein levels in (**a**) normalized to β-Actin and plotted relatively to si Ctr condition. Data are presented as mean values ± SEM of n = 4 independent experiments, *P < 0.05, **P < 0.01, ***P < 0.001, ns: non-significant (two-way ANOVA). **c** Quantification of the Oxygen Consumption Rate (OCR) per cell, measured by a Seahorse assay, in HCT116 cells treated with control (si Ctr), hnRNP U (si hnRNP U), DDX3X (si DDX3X) or a combination of hnRNP U and DDX3X (si hnRNP U + si DDX3X) siRNAs for 48 h. Data are presented as mean values ± SEM of n = 3 independent experiments. **d** Quantification of the Oxygen Consumption Rate (OCR) linked to basal and maximal mitochondrial respiration, ATP turnover, proton leak, and spare respiratory capacity, measured by a Seahorse

assay, in HCT116 treated with si Ctr, si hnRNP U, si DDX3X or a combination of si hnRNP U + si DDX3X for 48 h. Data are presented as mean values ± SEM of n = 3 independent experiments, *P < 0.05, **P < 0.01 (two-sided paired t-test). **e** Quantification of the ECAR linked to non-glycolytic acidification, glycolytic capacity and glycolytic reserve, measured by a Seahorse assay (glycolysis stress test), in HCT116 treated with si Ctr and si hnRNP U for 48 h. Data are presented as mean values ± SEM of n = 3 independent experiments, P-value = 0.0316 for the non-glycolytic acidification (one-sided paired t-test). **f** Proliferation assay in HCT116 cells treated with si Ctr, si hnRNP U#1, si hnRNP U#2 for 48 h. Data are presented as mean values ± SEM of n = 3 independent experiments, *P < 0.05, **P < 0.01 (two-sided paired t-test). For all the panels, Source Data is provided as a Source Data file indicating exact P-values for (**b**, **d**, **f**).

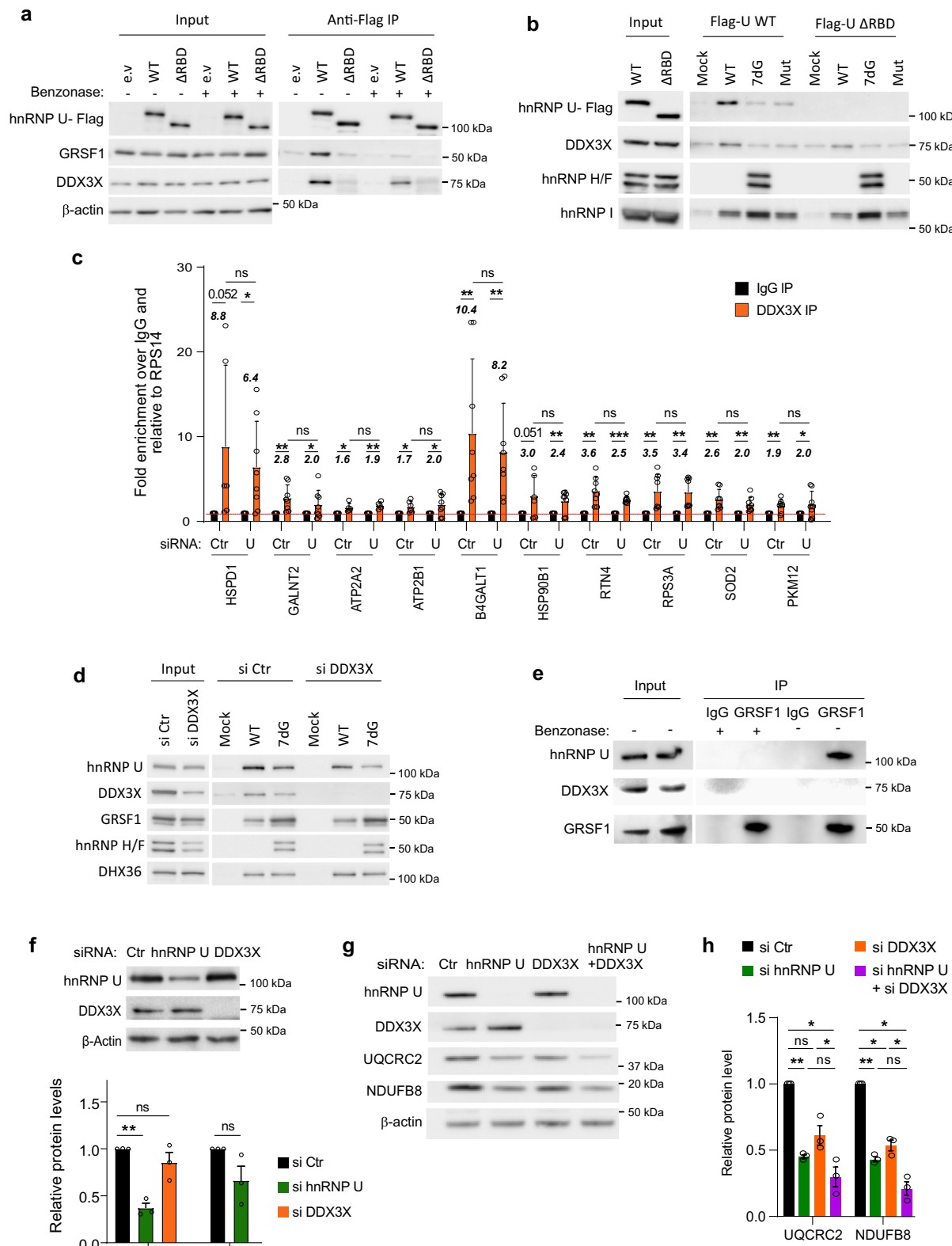

mitochondrial respiration. Although PhenDC3 or hnRNP U silencing mostly reduced mRNA translation and expression, we also observed examples of translational activation after these treatments (Fig. 2b, c). These different effects may be explained by the interaction between RG4s and other factors regulating translation or the competition between ligands and RBPs for RG4-binding[90].

The underlying translational regulatory mechanism, similar to the "bind, unfold, lock" model we proposed previously[26], could involve DDX3X recruiting hnRNP U to folded RG4s, which in turn facilitates their unfolding and makes GRSF1 binding accessible via hnRNP U and additional factors that remain to be characterized. Our results suggest that this RG4-dependent mechanism may act as an effector of

**Fig. 7 | hnRNP U cooperates with DDX3X and GRSF1 to regulate RG4-dependent translation. a** Immunoprecipitation (IP) with the Flag antibody of protein from cytoplasmic cell extracts (CE) of HCT116 cells transfected with flag-tagged unmodified hnRNP U (WT), hnRNP U RBD deleted (ΔRBD), or empty vector (e.v), followed by western blot analysis. **b** RNA affinity chromatography using the RG4 sequences either native (WT), 7-deaza-modified (7dG) or mutated (Mut) and CE of HCT116 cells transfected with plasmids encoding for hnRNP U-Flag WT or ΔRBD, followed by western blot analysis. **c** IP with the DDX3X antibody of cellulo RNA-protein complexes (RIP) in CE of HCT116 treated with control (Ctr) or hnRNP U (U) siRNAs, followed by RT-qPCR analysis. Fold changes are indicated in italic. *$P < 0.05$, **$P < 0.01$, ***$P < 0.001$ (one-sided paired t-test). **d** RNA affinity chromatography using the RG4 sequences either WT, 7dG, and CE of HCT116 treated control (si Ctr) or DDX3X (si DDX3X) siRNAs, followed by western blot analysis. **e** IP with the GRSF1

antibody of CE of HCT116 treated or not with benzonase, followed by western blot analysis. Shown is a representative result from n = 3 independent experiments. **f** Western blot analysis in HCT116 cells treated with si Ctr, si hnRNP U, or si DDX3X for 48 h and quantification of the protein levels normalized to β-Actin. $P$-value = 0.0067 for hnRNP U protein in si hnRNP U condition (two-sided paired t-test). **g** Western blot analysis in HCT116 cells treated with si Ctr, si hnRNP U, si DDX3X or both si hnRNP U + si DDX3X for 72 h. **h** Quantification of the protein levels in (**g**) normalized to β-Actin. *$P < 0.05$, **$P < 0.01$, (two-sided paired t-test). In (**a**, **b**, **c**, **d**, **f**, **g**, **h**), source data are provided as a Source Data file indicating exact $P$-values for (**c**, **h**). Data are presented as mean values ± SEM of n = 3 for (**f**, **h**), of n = 4 in duplicates for (**c**) independent experiments. In (**a**, **b**, **d**, **f**, **g**), the blot shown is a representative result from n = 3 independent experiments. When indicated, ns non-significant.

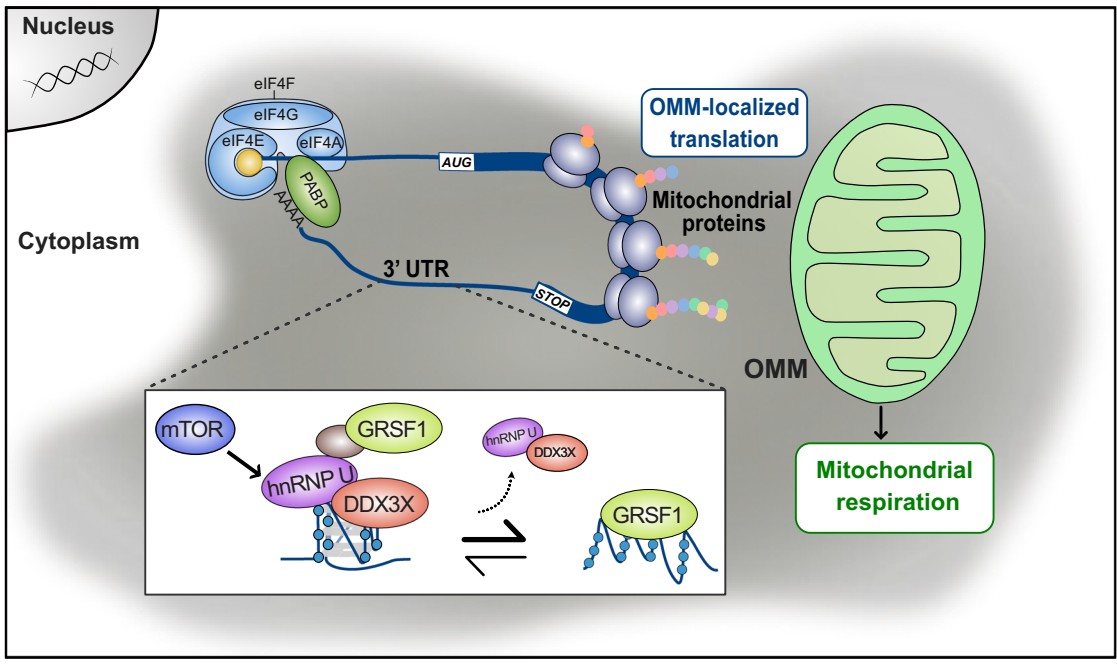

**Fig. 8 | Model for the role of hnRNP U-RG4 interactions in regulating OMM-localized mRNA translation of nuclear-encoded mitochondrial mRNAs.** Following the "bind, unfold, lock" model previously proposed[26], hnRNP U in complex with DDX3X unfolds RG4s and indirectly recruits GRSF1 onto G-rich sequences. This molecular mechanism, which is regulated by mTOR, activates OMM-localized

mRNA translation of nuclear-encoded mitochondrial mRNAs carrying RG4s in their 3′UTR, resulting in mitochondrial protein expression. Ultimately, this modulates mitochondrial functions, increasing energy metabolism and consequently cancer cell proliferation.

oncogenic mTOR signaling, which is known to regulate mitochondrial mRNA translation and thereby provide a source of ATP for protein synthesis[75]. Although the link between RG4s and upstream oncogenic signaling requires further investigation, our results showing that mTOR inhibition induces global folding of RG4s while reducing the binding of hnRNP U to RG4s and polysomes (Fig. 5h, i, Supplementary Fig. 8a, c) indicate the possibility that mTOR inhibition induces a reduction in ATP synthesis via inhibition of RG4/hnRNP U-dependent mitochondrial mRNA translation and that this ATP shortage in turn impacts on the activity of RG4 helicases, including DDX3X. This hypothesis is consistent with previous data showing that depletion of cellular ATP levels by inhibition of oxidative phosphorylation or glycolysis prevented RG4 unfolding during recovery from starvation-induced stress[77]. One important question that needs to be addressed, concerns the mechanism by which RG4s at 3′UTRs regulate mRNA translation. One possibility arising from recent ribosome profiling data (based on sequencing of mRNA fragments enclosed within the translating ribosome), which shows that the translation of the main ORF (mORF) is regulated by small ORFs located within 3′UTRs (dORFs)[91,92], is that RG4s could be involved in this regulatory mechanism, for which

the molecular details are still largely unexplored. Supporting this hypothesis, RG4s colocalizing with small ORFs located upstream of the ORF (uORF) were shown to regulate mORF translation[93]. In line with this possibility, we found that mitochondria-related terms were enriched among the mRNAs in which dORFs were experimentally identified[91,92]. Moreover, these elements are present in one-fourth of the RG4-containing mRNAs bound by hnRNP U (Supplementary Fig.11), indicating that RG4s might be involved in dORF-mediated regulation of mORF translation. Since RG4 stabilization, induced by ligands or by depleting hnRNP U, inhibits mitochondrial respiration and glycolysis in different cancer cell types (Figs. 1c, e, and. 6c–e, Supplementary Fig. 2c, d, Supplementary Fig. 4b, c, Supplementary 9g), the RG4/hnRNP U-dependent mechanism described here is not expected to be cancer type-specific per se. However, the quantitatively different effects of the ligand on mitochondrial respiration, and proportionally on the proliferation of the panel of tested cancer cells, correspond to the different level of dependency of those cells on mitochondrial function[2]. This is consistent with the view that cancer cells and tumors have heterogeneous metabolic preferences and dependencies[94]. Of note, the expression of hnRNP U is globally increased in cancer patient

samples compared to normal[95] and its expression is significantly associated with DDX3X in all types of TCGA cancers (R = 0,67; *P*-value = 0) (Supplementary Fig. 12a). Moreover, although the role of hnRNP U bound to RG4s deserves further investigation to understand its impact on patient outcome, hnRNP U and its direct targets show a pan-cancer prognostic significance (Supplementary Fig. 12b), suggesting that this regulation could have clinical relevance. The inhibition of mitochondrial translation and function has been explored as a therapeutic strategy for the treatment of cancer[8–10]. Furthermore, the clinical applicability of G4s as anticancer drugs is currently investigated in clinical trials[96]. Therefore, our observation that G4 ligands modulate cell proliferation, mitochondrial protein expression and function in cancer cells (Figs. 1,2, Supplementary Fig. 4) and that these effects are more pronounced in cancer cells than in normal cells (Supplementary Fig. 12c, d) could open up a whole new avenue of knowledge and therapeutic opportunities. Although the effects of altered RG4/hnRNP U function on glycolysis need to be further investigated, the ability to simultaneously target mitochondrial and glycolytic bioenergetics, thereby inducing a hypometabolic state, is of great interest for future dual-targeting strategies that could prevent compensatory effects between the two energy metabolism pathways[94].

## Methods

### Cell culture and treatment

HCT116 (ATCC CCL-247), HeLa(ATCC-CCL-2), HEK293T (ECACC #12022001) and U251 (ECACC #09063001) cells were grown in DMEM 4.5 g/L glucose supplemented with 10% FBS, 2 mM L-glutamine, 100 U/mL penicillin and 100 µg/mL streptomycin. SKMEL28 (ATCC HTB-68™) was maintained in RPMI 1640 supplemented with GlutaMAX, 10% FBS, 100 U/mL penicillin, and 100 µg/mL streptomycin. MCF7 (ATCC HTB-22) cells were grown in DMEM/F12 GlutaMAX supplemented with 10% FBS, 1 mM sodium pyruvate, 50 µg/mL gentamycin. MCF10A (ATCC CRL-10317) cells were maintained in DMEM/F12 GlutaMAX supplemented with 1 mM sodium pyruvate, 50 µg/mL gentamycine, 7 ng/mL of hEGF, 100 ng/mL of Cholera Toxin from *Vibrio cholerae*, and 1X Mammary Epithelial Growth Supplement (MEGS, Invitrogen S-015-5). Mycoplasma contamination was frequently assessed by PCR. Where indicated, cells were incubated at 37 °C with: PhenDC3 (Sigma SML2298), cPDS (Sigma SML1176), puromycin (Sigma P8833), cycloheximide (Sigma C7698), Torin 1 (Selleck Chemicals, S2827), Bafilomycin A1 (InVivoGen, tlrl-baf1), with the indicated concentration and for the indicated time.

### Plasmid construction

For generating hnRNP U- WT- Flag mutated in the siRNA (#1) site, a splicing polymerase chain reaction (PCR) was performed between two fragments generated by PCR using the couple of primers hnRNP U-FW/Mut-siRNA-RV and Mut-siRNA-FW/hnRNP U-RV-pcDNA3 and using pcDNA3-SAF-A-FLAG[97] as a matrix. The resulting hnRNP U WT -FLAG mutated in the siRNA site, was subcloned into pcDNA3 plasmid between BamHI and NotI. For generating hnRNP U- ΔRBD- Flag mutated in the siRNA (#1) site, a splicing PCR was performed between two fragments generated by polymerase chain reaction (PCR) using the couple of primers hnRNP U-FW/Mut-siRNA-RV and Mut-siRNA-FW/hnRNP U-RV-pEGFP and using pEGFP-SAF-A-dRBD-FLAG[98] as a matrix. The resulting hnRNP U- ΔRBD- Flag mutated in the siRNA site, was subcloned into pcDNA3 plasmid between BamHI and NotI. A list of primers used in the study is provided in Supplementary Data 2. All DNA constructs were confirmed mutation-free as tested by DNA sequencing.

### Cell transfection

siRNAs were transfected using the Lipofectamine RNAiMAX (Life Technologies) according to the manufacturer's instructions. In brief, cells were reverse-transfected with 2 nM of siRNA control (5′-GGUCCGGCUCCCCCAAAUG dTdT-3′) or 2 nM siRNA against hnRNP U (1: 5′-GUAGAACUCUCGUAUGCUA dTdT-3′; 2: 5′-GCGAAAAGCUGUU GUAGUU dTdT-3′), 10 nM siRNA against DDX3X (5′-CCUAGACCU GAACUCUUCAGAUAAU dTdT-3′) or 20 nM siRNA against GRSF1 (5′-GUGCCUCUCUGCUGCCGCAdTdT-3′) which were synthesized by Sigma. Cells were subsequently incubated at 37 °C for 48 h before harvesting and analysis. Media was changed 24 h after the transfection. Plasmids and RLuc-RNA reporters were transfected using the Lipofectamine 3000 (Life Technologies) according to the manufacturer's instructions. In brief, cells were transfected with 0.1 µg/cm² of plasmids encoding for hnRNP U WT- Flag or hnRNP U ΔRBD-Flag, with 50 ng/cm² of capped RLuc-B4GALT1 WT/Mut or RLuc-GALNT2 WT/Mut, and with 20 ng/cm² of capped Fluc-RNA (control). Cells were subsequently incubated at 37 °C for 16, 24, or 48 h before harvesting and analysis.

### RT-qPCR

Reverse transcription (RT) was performed on 500 ng total RNA (quantified with the Clariostar BMG and software v.5.40 R2, Labtech, and MARS Clariostar Analysis Software v.3.31) using the RevertAidH Minus First (Thermo Fisher Scientific EP0452) according to the manufacturer's protocol. qPCR analysis of cDNA (12.5 ng) was performed with the SybrGreen (KAPA KK4605) using the StepOne Applied Biosystems software. Expression of indicated mRNAs was normalized to the HPRT or RPLP0 as a reference. Relative levels of expression were quantified by calculating $2^{\Delta\Delta CT}$, where $\Delta\Delta CT$ is the difference in CT (cycle number at which the amount of amplified target reaches a fixed threshold) between the target and reference. All primer sequences are available in Supplementary Data 2.

### Immunoprecipitation

For RNA analysis, cytoplasmic (cytosolic + microsomal fractions (as described in "cell fractionation" of Methods section) extracts from HCT116 cells crosslinked using 1% formaldehyde/PBS for 10 min at 25 °C that is then quenched with 0.125 M glycine for 5 min, were digested with Benzonase (Millipore 70746-3) and DNase I (Thermo Fisher Scientific EN0521) for 1 h at room temperature. The digested cytoplasmic extracts were precleared on protein-sepharose beads for 1 h at 4 °C. BG4 (0.5 µg, expressed from the pSANG10-3F-BG4 plasmid (Addgene 55756), kindly provided by S. Balasubramanian), or hnRNP U (2 µg, Santa Cruz sc-32315) antibodies were incubated with 20 µL of slurry beads (washed and equilibrated in cell lysis buffer) for 1 h at 4 °C. 1 mg of cytoplasmic cell extracts was added on beads and incubated on a wheel overnight at 4 °C. After five washes of the beads with cell lysis buffer, the immunoprecipitated proteins and RNAs were eluted in the NT2 buffer (50 mM Tris pH 7.4, 1 mM MgCl₂, 0.05% NP-40). Subsequently eluted samples from the immunoprecipitation were treated with proteinase K (Euromedex EU0090-A). RNAs were extracted with Phenol/Chloroform and resuspended in 10 µL of water. 4 µL was reverse transcribed using the RevertAidH Minus First (Thermo Fisher Scientific EP0452) following the manufacturer's indication. Then, a 1/5 dilution of cDNA was analyzed by qPCR with the SybrGreen (KAPA, KK4605). The mRNA levels contained in these mRNP complexes were standardized against HPRT or RPLP0 mRNA levels (used as a reference) and compared to the RNA levels in the IgG control and input sample. For protein complexes analysis, cytoplasmic (cytosolic + microsomal fractions (as described in "cell fractionation" of "Methods" section) cell extracts were digested with Benzonase (Millipore 70746-3) and precleared with protein-sepharose beads for 1 h at room temperature. hnRNP U (2 µg, Santa Cruz sc-32315) antibody was incubated on a wheel overnight at 4 °C with 40 µL of protein-sepharose beads and 600 µg of precleared cytoplasmic cell extracts. After three washes of the beads with the cell lysis buffer and one last wash with buffer containing 10 mM Tris pH 8.0, 1.5 mM MgCl₂ and 140 mM NaCl, the

immunoprecipitated proteins were either resuspended in 60 μL elution buffer (0.5 M Tris pH 6.8, 40% glycerol and 8% SDS) for mass spectrometry analysis or 60 μL of 2X Laemmli buffer western blot analysis.

### In vitro transcription

All RNAs were obtained using the MEGAscript Kit (Invitrogen AM1333) as per manufacturer's instructions. 7.5 mM ATP/CTP, 6.75 mM UTP, 0.75 mM biotinylated UTP (Biotin-16-UTP, Lucigen BU6105H), and either 7.5 mM GTP or 6.75 mM 7-deazaguanine (TriLink N-1044) plus 0.75 mM GTP was used. B4GALT1 and GALNT2 DNA templates were generated through splicing PCR using the primers FW / RV or mut-FW/mut-RV (for the insertion of mutations into the RG4 sequence) and HCT116 cDNA (reverse transcription product) as matrix. For RNAs used in RNA chromatography experiments, the DNA templates were generated by annealing G3A2 WT, G3A2 Mut, NRAS WT, and NRAS Mut oligonucleotides. They were then cloned in the pSC-B-amp/kan plasmid from the Stratalone Blunt PCR cloning kit, then digested by NheI and purified. All oligonucleotide sequences are available in the Supplementary Data 3. For Rluc-RNA reporter assays, RNAs were capped using the Vaccinia Capping System (BioLabs) according to the manufacturer's instruction. RNA concentration was determined using the Clariostar BMG and software v.5.21 R4, Labtech, and MARS Clariostar Analysis Software v.3.20 R2.

### In cellulo RLuc-RNA reporter assay

In cellulo Rluc-RNA reporter assays were performed using the Dual-Luciferase Reporter Assay System Kit (Promega) according to the manufacturer's instruction. In brief, cells transfected with capped Rluc-RNA reporters and co-transfected with capped Fluc-RNA were lysed in the provided lysis buffer. Fluc and Rluc activities in cells were measured using a CLARIOstar Plus (BMG Labtech).

### Cell fractionation

Cells were gently resuspended in 500 μL of hypotonic lysis buffer (10 mM Tris pH 8.0, 10 mM NaCl, 1.5 mM MgCl₂, 1 mM DTT) and vortexed for 4 s. After centrifugation at $1000 \times g$ (4 °C) for 5 min, supernatant (cytosolic fraction) was recovered. Pellet fraction (washed twice with hypotonic lysis buffer) was resuspended in 500 μL lysis buffer A (10 mM Tris pH 8.0, 140 mM NaCl, 1.5 mM MgCl₂, 1 mM DTT, 0.5% NP40) and centrifuged at $1000 \times g$ (4 °C) for 5 min. The supernatant (microsomal fraction) was recovered. Pellet-nuclear fraction (washed twice with lysis buffer A) was resuspended in 500 μL of lysis buffer A with 50 μL of detergent mix (3.3% (w/v) sodium deoxycholate, 6.6% (v/v) Tween 40). After slow vortexing and incubation on ice for 5 min, the supernatant post-nuclear fraction was recovered (perinuclear fraction). The pellet-nuclear fraction (washed with buffer A) was resuspended in 500 μL of lysis buffer A supplemented with 0.1% SDS and sonicated. After centrifugation at $1000 \times g$ (4 °C) for 5 min, supernatant (nuclear fraction) was transferred into a clean tube.

### Mitochondrial purification

Approximately $40 \times 10^6$ cells were resuspended in 1 mL of MIB buffer (200 mM mannitol, 70 mM sucrose, 0.5 mM EGTA, 10 mM HEPES pH 7.5, 0.2% BSA, 10 mM NaF, 2 mM NaPyroP supplemented with 10 μL/mL of Protease Cocktail Inhibitor (Sigma, P8340)) and rested on ice for 10 min. 10 μL of the resuspended cells were saved as the total fraction. The rest of the extract was transferred to a 5-mL Potter-Elvehjem homogenizer with a Teflon pestle and the cells were broken with 50 strokes of the pestle on ice. Nuclei and unbroken cells were pelleted by centrifugation at $600 \times g$ at 4 °C for 10 min. The supernatant was collected in a new tube and the pellet was resuspended in 750 μL of MIB buffer and lysed again using the 5-mL Potter-Elvehjem homogenizer. After centrifugation at $600 \times g$ at 4 °C for 10 min, the supernatant was collected in a clean tube. This step was repeated once and

the pellet was then discarded. The supernatants collected from the three homogenization steps were centrifuged at $10,000 \times g$ at 4 °C for 10 min. The supernatant (cytosolic and microsomal fraction) was transferred to a new tube. The pellets (containing mitochondria) were pooled and washed 4 times in 1 mL of MIB buffer (the mitochondrial suspension was centrifuged at $10,000 \times g$ at 4 °C for 10 min for the first 2 washes and at $16,000 \times g$ at 4 °C for 2 min for the 2 following washing steps). The pellet was then resuspended in 1 mL of MIB buffer. 10 μl of the lysate was saved as the mitochondrial fraction. For OMM (Outer Mitochondrial Membrane) and IM/M (Inner Membrane and Matrix) purification, digitonin (Sigma, D141-100MG) was then added to the final concentration of 5 mg/mL, the mitochondrial suspension was mixed intensively and incubated for 15 min on ice. After centrifugation at $10,000 \times g$ at 4 °C for 10 min, the supernatant (OMM) was collected and the pellet (IMM) was resuspended in 1 mL of MIB.

### Proteinase K assay

After mitochondrial purification, pellet was resuspended in 400 μL of MIB without protease inhibitor. 100 μL of this mitochondrial suspension were distributed into 4 tubes and were incubated at 30 °C for 30 min with either 1% (w/v) Triton X-100 (Sigma, T8787) or 50 μg/mL proteinase K (Euromedex EU0090-A). The reaction was stopped by adding 2 μL phenylmethanesulfonyl fluoride solution at 200 μM for 10 min on ice.

### Western blot and antibodies

For immunoblotting analysis, proteins were resolved on 12.5 or 10% denaturing polyacrylamide gels and were transferred to nitrocellulose membranes. The blots were blocked for 30 min with TBST-5% milk and then probed overnight with primary antibodies against hnRNP U (1:1000, Santa Cruz sc-32315), β-Actin (1:2000, Sigma A5441), B4GALT1 (1:1000, GeneTex GTX131598-S), SOD2 (1:1000, Santa Cruz sc-133134), GALNT2 (1:1000, Sigma HPA011222), PKM (1:1000, Santa Cruz sc-365684), COX1 (1:1000, GeneTex GTX11619), COX4 (1:1000, Santa Cruz sc-376731), Flag (1:1000, Sigma F1804), DHX9 (1:1000, Abcam ab54593), hnRNP H/F (1:1000, Abcam Ab10689), PTB (1:1000, ATCC HB-94, clone BB7.7), MRPS34 (1:1000, Proteintech 15166), RPL28 (1:1000, Elabscience AB 32798), MRPL30 (1:1000, Abclonal A16529), DDX5 (1:1000, Abcam ab10261), PRMT1 (1:1000, cell signaling 2449S), TOMM20 (1:1000, GeneTex GTX133756), DHX30 (1:1000, Abcam 85687), GRSF1 (1:1000, Sigma HPA036985), OXPHOS (1:1000, Abcam ab110411), DDX3X (1:1000, Santa Cruz sc-365768), RPS6 (1:1000, Santa Cruz sc-74459), Tubulin (1:1000, Cell Signaling CS2146), Lamin A (1:1000, Santa Cruz sc-20680), Puromycin (1:1000, Millipore, MABE343), LARP1 (1:1000, Bethyl A302-087A), P(S240-244)-RPS6 (1:1000, Cell signaling 2215S), GRSF1 (1:1000, Sigma HPA036985) as well as secondary Anti-Rabbit (1:5000, Ozyme 7074S) and Anti-Mouse (1:5000, Ozyme 7076S) IgGs. The blots were developed using the ECL system (Amersham Pharmacia Biotech) according to the manufacturer's directions and images were quantified using FIJI software.

### Mass spectrometry bottom-up experiments

Number of samples was 10 with cPDS treated cells compared to untreated cells for total proteome analysis with n = 5 independent experiments; and 8 samples for IgG (Invitrogen # 14-4714-81) control or hnRNP U immunoprecipitation analysis with n = 4 independent experiments. Tryptic peptides were obtained by S-trap Micro Spin Column according to the manufacturer's protocol (Protifi, NY, USA). Briefly, proteins from either total proteome analysis (30 μg lysate) or hnRNP U immunoprecipitation eluates, were treated using a single step reducing and alkylating reaction in 200 mM TEAB (triethylammonium bicarbonate), pH 8.5, 2% SDS, 10 mM TCEP (tris(2-carboxyethyl)phosphine), 50 mM chloroacetamide in milliQ-grade H2O (mQ H2O) for 5 min at 95 °C. Denatured proteins were enzymatically digested for 14 h at 37 °C with 1 μg of sequencing-grade Trypsin (V511A,

Promega). After speed-vaccum drying, total proteome samples were solubilized in 2% trifluoroacetic acid (TFA) to undergo further peptide subfractionation by strong cationic exchange (SCX) StageTips, mainly as described[99]. Briefly, samples were loaded on SCX stage tips after hydratation and activation of membrane with acetonitrile (ACN) and 0.1% TFA respectively. Peptides were then washed using 0.1%TFA and eluted in 5 steps with 20% ACN, 0.5% formic acid, 75 mM or 125 mM or 200 mM, or 300 mM ammonium acetate for fraction 1 to 4 respectively, and 5% ammonium hydroxide, 80% ACN for fraction 5. Fractions were then dried again. All samples and fractions were resuspended with 10 µl of 10% ACN, 0.1% TFA in mQ H2O.

**Total proteome sample analysis**
LC-MS analyses were performed on a Dionex U3000 HPLC nanoflow chromatographic system (Thermo Fischer Scientific, Les Ulis, France) coupled to a TIMS-TOF Pro mass spectrometer (Bruker Daltonik GmbH, Bremen, Germany). One microliter of each fraction was loaded, concentrated, and washed for 3 min on a C18 reverse phase precolumn (3 µm particle size, 100 Å pore size, 75 µm inner diameter, 2 cm length, (Thermo). Peptides were separated on an Aurora C18 reverse phase resin (1.6 µm particle size, 100 Å pore size, 75 µm inner diameter, 25 cm length (IonOpticks, Middle Camberwell Australia) mounted onto the Captive nanoSpray Ionisation module, with a 120 min overall run-time gradient ranging from 99% of solvent A containing 0.1% formic acid in mQ H2O to 40% of solvent B containing 80% ACN, 0.085% formic acid in mQ H2O. The mass spectrometer acquired data throughout the elution process and operated in DDA PASEF mode with a 1.1 s/cycle, with Timed Ion Mobility Spectrometry (TIMS) enabled and a data-dependent scheme with full MS scans in Parallel Accumulation and Serial Fragmentation (PASEF). This enabled a recurrent loop analysis of a maximum of up to 120 most intense nLC-eluting peptides which were CID-fragmented between each full scan every 1.1 s. Ion accumulation and ramp time in the dual TIMS analyzer was set to 166 ms each and the ion mobility range was set from $1/K0 = 0.6$ Vs cm$^{-2}$ to 1.6 Vs cm$^{-2}$. Precursor ions for MS/MS analysis were isolated in positive polarity with PASEF in the 100–1.700 m/z range by synchronizing quadrupole switching events with the precursor elution profile from the TIMS device. The cycle duty time was set to 100%, accommodating as many MSMS in the PASEF frame as possible. Singly charged precursor ions were excluded from the TIMS stage by tuning the TIMS using the otof control software, (Bruker). Precursors for MS/MS were picked from an intensity threshold of 2.500 arbitrary units (a.u.) and re-fragmented and summed until reaching a "target value" of 20.000 a.u. while allowing a dynamic exclusion of 0.40 s elution gap. The mass spectrometry data were analyzed using Maxquant version 2.0.1.0[100]. The database used was a concatenation of Human sequences from the Uniprot-Swissprot reviewed database (release 2020-03) and the list of contaminant sequences from Maxquant. Cysteine carbamidomethylation was set as constant modification and acetylation of protein N-terminus and oxidation of methionine were set as variable modifications. Number of missed cleavages was set to 2, mass tolerance for precursor was 10 ppm and fragments 40 ppm, and minimum peptide length was 7. Second peptide search, normalization, and the "match between runs" (MBR) options were allowed. False discovery rate (FDR) was kept below 1% on both peptides and proteins. Label-free protein quantification (LFQ) was done using both unique and razor peptides with at least 2 such peptides required for LFQ. Statistical analysis and data comparison were done using the Perseus 1.6.15.0 software[101]. Student tests were done for proteins showing 70% of valid values in at least one condition.

**Immunoprecipitated sample analysis**
LC-MS analyses were performed on a Dionex U3000 HPLC nanoflow chromatographic system (Thermo) coupled to an Orbitrap Fusion Mass Spectrometer (Thermo). 1 µL of resuspended peptides was loaded, concentrated, and washed for 3 min on a C18 reverse phase precolumn (3 µm particle size, 100 Å pore size, 75 µm inner diameter, 2 cm length (Thermo). Chromatographic separation was performed on a 2 µm particle size, 100 Angström pore size, 75 µm internal diameter, 25 cm length C18 reverse-phase analytical column (Thermo) with a 1 h binary gradient from 99% solution A to 40% solution B. MS detection and data acquisition was performed throughout the elution process in a data-dependent scheme (top speed mode in 5 s) with full MS scans acquired with the orbitrap detector, followed by HCD peptide fragmentation and Ion trap fragment detection of the most abundant ions detected in the MS scan. Mass spectrometer settings for full scan MS were: 2E5 minimum intensity for AGC, 60,000 as target resolution, 350–1500 m/z as detection range, maximum ion injection time (MIIT) of 60 ms. HCD MS/MS fragmentation was permitted for 2–7+ precursor ions reaching more than 5.0E3 minimum intensity. Quadrupole-filtered precursors within 1.6 m/z isolation window were fragmented with a Normalised Collision Energy setting at 30. 2.0E4 AGC Target and 100 ms MIIT were the limiting ions accumulation values. A 30 s dynamic exclusion time was set. The mass spectrometry data were analyzed using Maxquant version 1.6.4.0. The database used was a concatenation of Human sequences from the Uniprot-Swissprot reviewed database (release 2019-10) and the list of contaminant sequences from Maxquant. Cysteine carbamidomethylation was set as constant modification and acetylation of protein N-terminus and oxidation of methionine were set as variable modifications. Number of missed cleavages was set to 2, mass tolerance for precursor was 4.5 ppm and fragments 0.5 Da and minimum peptides length was 7. Second peptide search was allowed while the "match between runs" (MBR) and the normalization options were not allowed. False discovery rate (FDR) was kept below 1% on both peptides and proteins. Label-free protein quantification (LFQ) was done using both unique and razor peptides with at least 2 such peptides required for LFQ. Statistical analysis and data comparison were done using the Reprint software.

**Seahorse assay**
Cells were plated at a density of $2 \times 10^4$ HeLa, $2.5 \times 10^4$ HCT116, and $1.8 \times 10^4$ SKMEL28 per well in Agilent Seahorse XF24 cell culture microplates (Agilent, 100777-004), reverse-transfected with 2 nM of siRNAs when indicated, and incubated at 37 °C for 48 h before Seahorse assay. The day before the assay, the sensor cartridge was placed into the calibration buffer medium (Agilent, 100840-000) to hydrate at 37 °C without CO2 overnight. When indicated, cells were treated with PhenDC3 or cPDS overnight. Cells were incubated in 500 µL Seahorse XF Base Medium Minimal DMEM without Phenol Red (Agilent, 103335-100) 1 h at 37 °C in CO2 free-atmosphere. Media was supplemented with 1 mM sodium pyruvate, 2 mM L-glutamine, and 10 mM glucose for Mito Stress Test, and with 1 mM sodium pyruvate, 2 mM L-glutamine only for the Glycolysis Stress Test. Drugs were loaded into the cartridge and were used in the following final concentrations in Seahorse media: oligomycin (2 µM), carbonyl cyanide-4-(trifluoromethoxy)phenylhydrazone (4 µM for HeLa and 2 µM for HCT116 and SKMEL28), antimycin (1 µM), rotenone (1 µM), glucose (10 mM) and 2-DG (50 mM). Real-time oxygen consumption rate (OCR) and extracellular acidification rates (ECAR) were measured using a Seahorse XF24 Extracellular Flux Analyzer (Agilent Technologies, Santa Clara, CA, USA) setting on Mito or Glycolysis Stress Test program. Data were analyzed with Wave Desktop 2.6.1 software. Respiratory measurements were normalized by the quantity of proteins. For this, the wells were lysed in buffer containing 10 mM Tris pH 8.0, 150 mM NaCl, and 0.5% NP-40, and proteins were quantified by BCA assay.

**Flow cytometry**
For detection of mitochondrial mass and membrane potential, cells were labeled with 100 nM MTG (Thermo Fisher Scientific, 11589106)

and 200 nM TMRE (Sigma, T669), respectively, and incubated at 37 °C for 20 min. Then cells were collected by trypsinization and 100,000 cells by condition were transferred in 96-well plate. Cells were washed with 1X PBS and resuspended in 150 μL of Annexin-binding buffer (10 mM HEPES pH 7.4, 140 mM NaCl, 2.5 mM CaCl$_2$) containing 1 μL Annexin V-BV421 antibody (BD Biosciences, 563973). The fluorescent intensity of 10,000 cells was detected by flow cytometry on a FACS instrument (MACSQuant®VYB, Miltenyi Biotec, San Diego, CA, USA). Data were analyzed using FlowJo software (FlowJo LLC, Ashland, OR, USA).

## BioCyTASQ labelling and imaging
Square coverslips (Thermo Fisher Scientific, 10474379) were heat-sterilized before cell seeding. HeLa cells were seeded at a density of $1 \times 10^5$ cells per coverslip in 6-well plates and transfected with siRNAs when indicated. For BioCyTASQ labeling, 24 h after seeding, cells were incubated with 1 μM BioCyTASQ and, when indicated mitochondria were labeled by incubating live cells with MitoTracker Deep Red (Thermo Fisher Scientific, 12010156) in HBSS medium for 30 min at 37 °C then cells were washed once with 1X PBS. For RNase controls, cells were permeabilized by incubating them in CSK buffer (10 mM PIPES pH 7.0, 100 mM NaCl, 300 mM sucrose, 3 mM MgCl$_2$, 0.1% Triton X-100) supplemented or not with 30 μg/mL of RNaseA and 75 U/mL of RNase T1 (RNase A/T1 mix, Thermo Fisher Scientific EN0551). Cells were fixed and permeabilized with ice cold MeOH for 10 min at room temperature, washed with 1X PBS (3 × 5 min), then incubated for 1 h at 25 °C with IkB (Abcam, ab76429, 1:300 in 1% BSA/1X PBS) or TOMM20 (GTX, 133756, 1:300 in 1% BSA/1X PBS), followed by 1 h incubation at 25 °C in a light-tight box with Streptavidin-Cy3 (1 μg/mL, Thermo Fisher Scientific, 10768023) and fluorescent secondary antibody Alexa Fluor 488 donkey anti-mouse (Invitrogen, A21202, 1:800 in 1% BSA/1X PBS) or Alexa Fluor 647 goat anti-rabbit (Invitrogen, A21245, 1:300 in 1% BSA/1X PBS). Cells were washed with 1X PBS (3 × 5 min) and once with H$_2$O. Nuclei were labelled with 1 μM DAPI (Euromedex, 1050-A) alone, or for QUMA labeling with 1 μM of QUMA for 15 min at room temperature. Coverslips were mounted onto glass microscope slides with Anti-fade glass mountant (Thermo Fisher Scientific, 15898391). The cells were imaged with a Zeiss LSM 880 Fast Airyscan confocal microscope with 20× air and 40× oil objectives. Images were processed and quantified using ZEN Blue and ImageJ software.

## RNAscope
$1.5 \times 10^4$ HeLa cells were seeded in 8 wells chambered coverslip (Clin-isciences, 80827). When indicated, cells were reverse-transfected with siRNAs for 48 h or treated with 20 μM PhenDC3 24 h after seeding for 24 h at 37 °C or with 100 μg/mL puromycin for 1 h at 37 °C. Cells were washed with 1X PBS and fixed with 10% Neutral Buffered Formalin (NBF) at 37 °C for 30 min followed by protease digestion (RNAscope Protease III (Advanced Cell Diagnostics, ACD) diluted at 1/50 in 1X PBS) during 15 min at room temperature. Cells were then processed according to the RNAscope protocol using the RNAscope Fluorescent Multiplex Kit (ACD, 320851) and revealing the labeled probes of AKAP1, HSPD1, GPD2, or RLuc mRNA with Alexa Fluor 550 (ACD) and MTND5 mRNA with Alexa Fluor 647 (ACD). For immunofluorescence staining of AKAP1, HSPD1, or GPD2 protein, after the last step of hybridization, cells were washed and incubated with a blocking solution (5% bovine serum albumin (BSA) in 1X PBS) for 30 min at room temperature. Immunofluorescence was then performed using the AKAP1 antibody (Novus Biologicals, NBP2-15319, 1:200 in 1% BSA/1X PBS), HSPD1 (Cell signaling, 4869S, 1:200 in 1% BSA/1X PBS) and cells were incubated overnight at 4 °C. Immunolabeling were visualized the next day with fluorescent secondary antibody Alexa Fluor 488 goat anti-rabbit (Thermo Fisher Scientific, A11008, 1:1000 in 1% BSA/1X PBS). Cells were finally washed, and nuclei were labeled with DAPI (ACD) for 1 min. Cells were conserved in 1X PBS at 4 °C and high resolution fluorescence

microscopy images were taken with a Zeiss LSM 880 FAST Airyscan using a 63X oil objectives.

## Proximity ligation assay
For the proximity ligation assay, the Duolink kit was used (Sigma Duolink In Situ orange DUO92102) with Puromycin (Mouse Millipore MABE343: 1/10,000), AKAP1 (Rabbit bio-techne NBP2-15319 1/2000), IkB (Rabbit Abcam ab76429 1/500) or RLuc (Rabbit Invitrogen PA1180 1/200) primary antibodies. For the Puro-AKAP1 PLA, cells were seeded into 12-well plates containing sterilized coverslips. The next day, cells were treated with 20 μM PhenDC3 or DMSO for 24 h then treated with 100 μg/mL of Cycloheximide for 25 min followed by 3 μM of puromycin for 5 min. Cells were fixed with 4% PFA for 20 min and permeabilized with PBS 5% BSA containing 0.01% saponine for 1 h at RT. The rest of the assay was done as per the manufacturer's protocol. Cells were visualized at room temperature by using a high-resolution confocal microscope (Zeiss, LSM880). For the PLA with RLuc reporters, cells were seeded in 24-plates containing sterilized coverslips, and then transfected with 100 ng of either RLuc-GALNT2 WT, Mut, or 7dG reporters for 2 h prior CHX and puromycin treatment.

## Cell proliferation
HeLa, HCT116, SKMEL28, MCF7, and MCF10A cells reverse-transfected with siRNAs or treated with 20 μM PhenDC3 or 20 μM cPDS were harvested and counted with Cellometer Mini Automated Cell Counter (Nexcelom Bioscience, Ozyme) on day 4 to 7.

## RNA chromatography
HCT116 or U251 cytoplasmic (cytosolic + microsomal fractions (obtained as described in "Cell fractionation" of the Methods section)), mitochondrial or OMM fractions (both obtained as described in "Mitochondrial fractionation" of the "Methods" section) were pre-cleared with 20 μL of streptavidin acrylamide beads (Thermo Fisher Scientific) in the binding buffer containing 20 mM Tris pH 8.0, 1 mM DTT, 100 mM KCl, 0.2 mM EDTA for 1 h at 4 °C. For RG4 structuration, 1 μg of in vitro-transcribed biotinylated RNAs were boiled for 5 min in one volume of 1x PBS supplemented with 2 M KCl and cooled down at room temperature for 20 min. Biotinylated RNAs were then immobilized on 10 μL of streptavidin acrylamide beads by incubation in the binding buffer for 1 h at 4 °C. The RNA fixed on beads was then combined to 200 μg of precleared cytoplasmic or mitochondrial extracts or 30 μg of OMM extracts for 3 h at 4 °C. The beads were collected by centrifugation, washed five times with 1 mL of the binding buffer, resuspended in 30 μL of elution buffer (50 mM Tris pH 8.0, 1% SDS), and heated to 95 °C for 10 min. After centrifugation at max speed, the supernatant was collected and loaded onto an SDS–PAGE gel and analyzed by western blot.

## Polysomes
For a detailed protocol, see in ref. 102. Briefly, around $4 \times 10^7$ cells were treated with 100 μg/mL cycloheximide (CHX) for 5 min at 37 °C, washed twice with ice-cold 1X PBS supplemented with 100 μg/mL CHX (PBS/CHX), and scraped on ice in PBS/CHX. After centrifugation for 5 min at $200 \times g$, the cell pellet was gently resuspended in 500 μL of hypotonic lysis buffer (5 mM Tris pH 7.5, 2.5 mM MgCl$_2$, 1.5 mM KCl, 100 μg/mL CHX, 2 mM DTT, 0.5% Triton X-100, 0.5% sodium deoxycholate, 200 U/mL RNaseOUT (Invitrogen, 10777019), 2 mM phenylmethanesulfonyl fluoride, 1 mM sodium pyrophosphate, 1 mM sodium fluoride and 2 mM sodium orthovanadate) and vortexed for 5 s. After incubation on ice for 10 min, the lysate was centrifuged at $16,000 \times g$ for 7 min at 4 °C and a volume of supernatant corresponding to 20 OD260 nm was layered on a 11.3 mL continuous sucrose gradient (5–50% sucrose in 20 mM HEPES pH 7.6, 0.1 M KCl, 5 mM MgCl$_2$, 10 μg/mL CHX, 0.2 mM PMSF, 10 U/mL RNaseOUT). After 2 h of ultracentrifugation at $222,228\,g$ in a SW41-Ti rotor at 4 °C,

fractions were collected with an ISCO density gradient fractionation system (Foxy Jr fraction collector coupled to UA-6UV detector, Lincoln, NE). The settings were as follows: fraction time, 35 s/fraction; sensitivity of the OD254 recorder, 1. The absorbance at 254 nm was measured continuously as a function of gradient depth; 16 fractions of ~0.8 mL were collected. Equal amounts of RNA from each fraction were extracted by using TRIzol LS (Invitrogen, 10296028) and subjected to RT-qPCR analysis to determine the polysomal mRNA distribution. Protein from individual fractions was extracted by using isopropanol precipitation and analyzed by western blot.

## RNA-Seq
12 RNA libraries were prepared with the NextFlex Rapid Directional mRNA-Seq kit (PerkinElmer) with polyA+ mRNA enrichment, and sequenced on the ProfileXpert platform, on an Illumina Nextseq500 sequencing machine with a single read protocol using high ouput flowcell (75 bp; 30 M reads). ERCC spikes (Thermo Fisher Scientific) were added to samples to control all the procedures. Demultiplexing has been performed using Bcl2fastq v2.17.1.14 software generating Fastq files.

## In vitro translation of Rluc-RNA reporter
Capped Rluc-RNA reporters were pre-folded by heating at 65 °C for 2 min and slowly cooled at room temperature for 10 min. Pre-folded RNAs (100 ng) were incubated in RRL containing 10 μM amino acid mix minus Methionine, 10 μM amino acid mix minus Leucine, 1 mM RNAseOUT, with or without 50 ng of recombinant hnRNP U (Origene, catalog number TP301627) at 30 °C for 1 h 30 min. Capped Fluc-RNA (40 ng) was co-translated in each well as a control. Fluc and Rluc activities were measured using the Dual-Luciferase Reporter Assay System kit (Promega) according to the manufacturer's instructions.

## SUnSET
Cells were incubated with 1 μg/mL of puromycin for 10 min at 37 °C. For inhibition of global protein synthesis, cells were pretreated with 100 μg/mL CHX 5 min at 37 °C before the addition of puromycin. After two washes in ice-cold 1X PBS, cells were scraped on ice in 1X PBS, centrifuged at $200 \times g$ for 5 min and lysed in the lysis buffer (10 mM Tris pH 8.0, 150 mM NaCl, 10% glycerol, 1% NP-40, 2 mM phenylmethanesulfonyl fluoride, 1 mM sodium pyrophosphate, 1 mM sodium fluoride and 2 mM sodium orthovanadate). Puromycin incorporation was detected using western blot analysis.

## Isoelectric focusing
Cells were treated with 300 nM Torin 1 or a vehicle (DMSO) for 3 h at 37 °C. After two washes in ice-cold 1X PBS, cells were scraped on ice in 1X PBS and centrifuged at $200 \times g$ for 5 min. For lambda phosphatase (New England BioLabs, P0753S) treatment, cells were lysed in lysis buffer (10 mM Tris pH 8.0, 150 mM NaCl, 10% glycerol, 1% NP-40, 2 mM phenylmethanesulfonyl fluoride, 1 mM sodium pyrophosphate, 1 mM sodium fluoride and 2 mM sodium orthovanadate) and 200 μg of proteins were incubated with 1200 U/μL of lambda phosphatase for 1 h at 30 °C. Proteins were then precipitated using the 2D Clean Up Kit (GE Healthcare) according to the manufacturer's instructions. Protein pellet was resuspended in UTC buffer (8 M urea, 2 M thiourea, 4% CHAPS) and quantified using RC DC™ Protein Assay Kit (Bio-Rad). Sample with 80 μg proteins resuspended in rehydration buffer (8 M urea, 2 M thiourea, 2% CHAPS, 10 mM DTT, 1.2% IPG buffer pH 3-10, bromophenol blue) was applied onto IPG strip (7 cm, pH 3-10, Bio-Rad, 1632000) for 24 h. After the rehydration of the IPG strip, first dimension migration was performed using the Ettan™ IPGphor™ 3 IEF system (Cytiva) to separate protein according to their isoelectric point according to the following protocol: step and hold 300 V, 45 min; gradient 1000 V, 1 h; gradient 4000 V, 2 h 30; step and hold 4000 V, 20 min. A default temperature of 20 °C with a maximum current of

50 μA were used. The strip was equilibrated by incubating it twice for 15 min at room temperature in equilibration buffer 1 (6 M Urea, 2% SDS, 50 mM Tris pH 8.6, 30% glycerol, 10 mg/mL DTT, bromophenol blue) and then in equilibration buffer 2 (6 M Urea, 2% SDS, 50 mM Tris pH 8.6, 30% glycerol, 47 mg/mL iodoacetamide, bromophenol blue). Second dimension was performed on 10% SDS-PAGE gel to separate the proteins based on their molecular weight. Proteins were then transferred onto nitrocellulose membrane for 1 h 30 min at 400 mA. Membrane was blocked for 30 min with TBST-5% milk and proteins were monitored by an immunoblotting analysis.

## Data analysis
Data analyses were performed with Microsoft Excel, Graphpad Prism 9, ImageJ v 1.53, R v 3.6.1, RStudio v 1.0.153, Zen Blue 2.3, FlowJo 10.8.1, Wave Desktop 2.6.1, StepOne Applied Biosystems software v2.2.2, and figures were prepared with Affinity designer v 1.10.4.1198, Microsoft Powerpoint, Inkscape v 0.92.4 and Gimp v 2.10.18. Transcriptome-wide prediction of RG4s, performed on the Gencode Human GRCh38 v32 set of transcripts by means of QGRS Mapper (https://doi.org/10.1093/nar/gkl253) were obtained from the QUADRatlas database[39], and those predicted RG4s with score greater or equal to 21 were considered for further analysis. Densities per Kb/Mb were obtained by dividing the number of sites/RG4 by the total length of the corresponding genomic region obtained from the genome assembly annotations. Eventually, distance of CLIP-derived binding sites with respect to RG4 elements were computed using bedtools v2.25 (doi:10.1093/bioinformatics/btq033). Functional enrichment analysis on the Gene Ontology was performed in R with the topGO package (Alexa A, Rahnenfuhrer J (2018). topGO: Enrichment Analysis for Gene Ontology. R package version 2.44).

## CLIP data analysis
ENCODE eCLIP binding sites for human RBPs were obtained from the ENCODE Data Portal, retrieving the replicate-merged, IDR-assessed peaks for each available RBP in both cell lines. Binding sites annotation for gene and genomic region of origin was performed with ctk (10.1093/bioinformatics/btw653). Analysis of the HNRNP U cyto-CLIP dataset[61] was performed as follows. Reads were trimmed (minimum quality 30, minimum length 18 nt) and adapters removed with Trim Galore (https://github.com/FelixKrueger/TrimGalore). Reads were then aligned to the human genome hg38 assembly with STAR (10.1093/bioinformatics/bts635). PCR duplicates were collapsed, and peaks were eventually called with PIRANHA v 1.2.1 (doi: 10.1093/bioinformatics/bts569), using an FDR threshold of 0.05. Binding sites annotation for gene and genomic region of origin was eventually performed with ctk (10.1093/bioinformatics/btw653).

## RNA-seq data analysis
Reads were trimmed (minimum quality 30, minimum length 18 nt) and adapters removed with Trim Galore (https://github.com/FelixKrueger/TrimGalore). Reads were then aligned to the human genome hg38 assembly with STAR (10.1093/bioinformatics/bts635), obtaining raw gene counts based on the GENCODE v43 annotation. Differential expression was eventually computed with DESeq2[103].

## Statistical analysis
Statistical analysis of the difference between two set of data was assessed using one-sided or two-sided paired t-test (GraphPad Prism). Differences in enrichment between gene sets were assessed with a chi-square test for unequal proportions (R). P-values of less than 0.05 were considered to be significant (*, $P < 0.05$; **, $P < 0.01$; and ***, $P < 0.001$). Binding site density was statistically assessed for differences with respect to random sites by performing a 1000-sample bootstrap analysis, generating an equal number of binding sites sets, each chosen at random from the genome and composed of as many sites as the true

set. An empirical *P*-value was then calculated on the resulting binding sites densities, comparing the true set and the 1000 sets density.

## Reporting summary

Further information on research design is available in the Nature Portfolio Reporting Summary linked to this article.

## Data availability

Polysome sequencing data have been deposited in NCBI's Gene Expression Omnibus and are accessible through GEO Series accession number GSE239640. The mass spectrometry proteomics data have been deposited to the ProteomeXchange Consortium via the PRIDE partner repository with the dataset identifier PXD036780, PXD046236 and PXD046213 and are provided in the Supplementary Data 1. Source data are provided with this paper.

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

## Acknowledgements

We are grateful to people from SM laboratory—S. Bourdon, A. Desprairies, T. Hullo for assistance with some experiments, as well E. Espinos and S. Manenti—for discussions and advices; to members of E. Sarry's team at CRCT—L. Stuani, T. Farge, E. Boet—for discussion, analysis, design of experiments and materials; to members of the technology cluster at CRCT—L. Ligat—for helping with confocal microscopy analysis. We thank F. Guillonneau, V. Salnot et J. Bruce from the 3P5 proteomics facility of the Université de Paris. The authors would like to thank Severine Croze and Joel Lachuer, from « ProfileXpert»/Viroscan3D (service de « Genomique & Microgenomique », Université Lyon 1, SFR sante LYON-EST, UCBL-INSERM US 7-CNRS UMS 3453) for RNA-Seq experiments. We thank R. Flores-Flores from Phi platform of I2MC, Toulouse for pipeline analysis of RNA-scope images. This work was supported by institutional grants (INSERM, Université Toulouse III - Paul Sabatier, CNRS) and by funding from ARC (Association pour la Recherche Contre le Cancer, Programme labellisé Fondation ARC to S.M.), Université de Toulouse, LNCC (Ligue contre le cancer to S.M.), Emergence Cancéropôle GSO to S.M., and ANR (ANR-17-CE12- 0017-01 to A.C.). L.D. was supported by MENRT, FRM, and FON-ROGA, S.S. by LNCC, M.C. and Q.R. were supported by ARC, P.H. by ANR (ANR-17-CE12-0017-01).

## Author contributions

S.M. conceived the project. S.M. designed and supervised the experiments with significant contribution from A.C. and assistance by M.C from some experiments. L.D. performed most of the experiments, with assistance by S.S., Q.R., M.C., B.HS., P.H., and N.S.L. D.M. contributed by providing BioCyTASQ and assistance for its utilization. M.L. and M.L.G. performed proteomic analysis and statistical data treatment. E.D. performed bioinformatic analysis. S.M. wrote the manuscript with input from all the authors.

## Competing interests

The authors declare no competing interests.
