## [Transparent Peer Review file · Nature Communications]

RNA G-quadruplexes control mitochondria-localized mRNA translation and energy metabolism

Corresponding Author: Dr Stefania Millevoi

Version 0:

Reviewer comments:

Reviewer #1

(Remarks to the Author)

A manuscript by Dumas et al. describes a series of experiments that aim to reveal a link between G-quadruplex RNA formation and regulation of translation of nuclear-encoded mitochondrial proteins. The Authors performed a comprehensive analysis of publicly available transcriptomic data to identify mRNAs, the translation of which can be regulated by G4 formation. This led to the hypothesis that such a mechanism can contribute to the regulation of mitochondrial proteome. Overall, this is a well-written manuscript that describes technically sound experiments, and the manuscript is of general interest. However, the major conclusion that G4 formation regulates the translation of mitochondria-associated mRNAs needs direct evidence. The Authors show that stabilization of G4 structures, similarly to translation inhibition with puromycin, leads to decreased levels of tested AKAP1 protein. However, it does not prove that postulated G4 structures affect mitochondria-associated translation, in fact, the Authors describe a similar tendency for mitochondrial-encoded ND5 mRNA, which is not expected to form G4RNA. The Authors should map G4 motifs in a few selected mRNAs, change their sequence (or synthesized using 7-deaza G) to prevent G4 formation, and test their translation after mRNA transfection into cells.

Other comments

1. How the results of cell number/proliferation of HCT116 should be interpreted, seem to be contradictory (Fig. 1e and Fig. S1e)
2. Fig. 3i – cells are too small to be able to assess the staining
3. Why components of mitochondrial translation machinery co-IP with hnRNP U (Fig. 4f). hnRNP U is absent in the mitochondrial matrix.
4. The Authors suggest that hnRNP U regulates the translation of mRNA by affecting G4RNA. This should be confirmed with a reporter mRNA that does not form G4.
5. All RNAi-based experiments miss rescue controls (complementation of phenotype by RNAi-insensitive version of transgene/mRNA). Alternatively, the same phenotype should be demonstrated by CRISP-Cas9 knockout.
6. The Authors imply that their data suggest a link between cancer and regulation of mitochondrial proteins translation. However, all experiments are done using cancer cells. Either normal, primary cells will be used for comparison, or authors will remove/tone down their claims concerning cancer biology (still, the manuscript will be interesting for being considered for publication).

Reviewer #2

(Remarks to the Author)

HNRNPU is traditionally thought of as a nuclear restricted hnRNP protein. However, a number of large scale omic and more targeted studies indicate that it might also play a cytoplasmic role. Here the authors show that there is fraction of hnRNPU present in the cytoplasm, consistent with other studies. They develop the argument that RG4s can regulate translation, which is not novel and make the case that HNRNPU plays a role in this process, specifically regulating the translation of mitochondrial mRNAs through binding to the OMM. This later point would be of wide interest and novel if it were proven convincingly.

HNRNPU is an extremely abundant RNA binding protein which means it appears in many mass spectrometry lists of hits. Therefore, it is not surprising that it appeared at the intersection of the datasets they used to identify proteins which might regulate RG4 translation. At least two other proteins also appeared at this intersection (which are also abundant) yet these are not considered further. My concern is that HNRNPU is here mainly because of its abundance not the reasons the

authors are hoping for. Therefore the reasons for choosing HNRNPU to focus on are somewhat tenuous and would require a compelling set of functional experiments to confirm its role. I did not feel the manuscript had this compelling set of follow on functional experiments. I do not dispute the mitochondrial association of HNRNPU and the mass spectrometry showing the abundance of mitochondrial proteins is convincing but whether it is really regulating translation of RG4 RNAs is not adequately demonstrated I feel. My chief concern is that the functional experiments in vivo rely on siRNA treatment for 48 hours. HNRNPU plays a key role in pre-mRNA splicing, chromatin association of RNA and RNA stability and there is a considerable literature describing this. So how do we know that during the 48 hour siRNA treatment the effects on these other processes, which are dramatic and well documented, do not indirectly alter translational processes by affecting other proteins which play a direct role in this process? Overall the study relies too much on correlative data and lacks direct functional in vivo evidence for a role for HNRNPU in the proposed process. Often the effects reported are small e.g. Fig. 2d, Specific points:

1. In the abstract and introduction the authors refer to RNAs frequently when I think they mean mRNAs. Using the term RNA is misleading because there are many other types of RNA besides mRNA. When I first read the abstract I had the impression that they were talking about a non-coding RG4 RNA regulating translation not an RG4 embedded within an mRNA.
2. Looking at Fig 1A my impression is that the BioCyTASQ probe mainly lights up the cytoplasm where mitochondria are so without seeing the distribution of some general cytoplasmic marker not found in mitochondria as a control it is difficult to unequivocally arrive at the conclusion that RG4s map to mitochondria.
3. Whilst phenDC3 alters proliferation of cancer cells, is this effect specific to cancer cells? Without evidence to suggest that is the case it is not a particularly interesting result. Lots of compounds alter cell proliferation but only a more restricted group specifically alter the proliferation of cancer cells.
4. In fig. 2B it appears that some RG4 containing mRNAs are translated better and some worse following stabilisation of RG4s. Why the differential effects? This is not discussed and argues against a simple model whereby RG4s recruit proteins which unwind them and then drive translation of those mRNAs.
5. The data in Supp fig. 2C showing that perturbation of RG4 containing mRNAs has an impact on RG4-less mRNAs, establishes that indirect effects are at play in this system. This again reinforces how important it is to design experiments which directly rather indirectly test the hypotheses presented.
6. The effects presented in Fig 2c seem very small. It's hard to be persuaded these are direct effects which I would have expected to be more dramatic.
7. Fig. 2e,f. Again are these indirect effects since RG4 stabilisation no doubt has many effects on the cell? A more direct test would have been to engineer an mRNA to remove the RG4 region and demonstrate a direct effect on its mitochondrial localisation and translation.
8. Fig. 3g. Whilst this figure looks superficially compelling, what does this plot look like for other proteins. There are eCLIP datasets available for hundreds of protein in ENCODE so how HNRNPU specific is this distribution?
9. HNRNPU is an essential gene and we are told that in one set of experiments its knockdown by RNA has no effect on proliferation (Supp Fig 5e) yet later in the paper we see that there is indeed an effect on proliferation at later time points. I suspect proliferation is already affected at the 48 hour time point but perhaps the cells have not yet fully detached and died. This brings me on to the point that 48 hours of HNRNPU siRNA treatment provides ample opportunity to alter splicing of many pre-mRNAs and chromatin localisation of many RNAs and a myriad of other effects e.g. alterations to mRNA stability which HNRNPU also regulates. So how do we know that indirect effects on some other mRNA are not causing the effects on translation of RG4 containing mRNAs?
10. The effects of HNRNPU KD on AKAP1 mRNA translation (Fig. 5b) seem extremely small if they even exist yet this mRNA is repeatedly used as an example for follow on experiments e.g. Fig. 5d where again the effects are very modest. This data would have been more convincing if the RNA in polysomes had been subject to mRNA seq and a global view of the effects on translation presented. A single negative control mRNA ActB is presented which makes it hard to derive any statistical significance about whether RG4 containing mRNAs are preferentially altered in their translation following loss of HNRNPU.
11. Fig 3e. The affinity chromatography could be revealing an indirect interaction between hnRNPU and the RG4 via another protein. If hnRNPU selectively binds RG4s how does it do it? Via its RGG box? Some functional complementation using mutant forms of HNRNPU might have been useful to address this important mechanistic issue.

Reviewer #3

(Remarks to the Author)

The work by Dumas and colleagues proposes the role of RG4 in the regulation of mitochondrial activities via its co-localization and co-regulation of translation of mitochondria-localized protein synthesis. The work is important and original. It opens new directions to study in vivo functions of RG4s.

I have however have several major issues to be clarified before potential acceptance of this work for publication

1) Abstract suggests model that " whereby the RG4 folding dynamics, under the control of oncogenic signaling and modulated by small molecule ligands or RG4-binding proteins, modifies mitochondria-localized cytoplasmic protein synthesis". I do not think that provided data supports such model directly

2) Connection to cell signaling, especially to oncogenic signaling, is weak. Data based on Fig 5e and Sup 5f-g is not sufficient for such statement.

3) Authors have not used any rescue experiments. This is huge minus. All depletion experiments should be accompanied by

rescue.

4) Experiments using G4 ligands are necessary but pleiotropic in nature. Experiments with phenDC3 should be repeated using other ligand (Fig 2).

5) If effects of Torin 1 are very specific to the hnRNP U-mediated regulation, would inhibition of mTOR result in the similar effects as observed with puromycin and phenDC3 (Fig 2C-f)?

6) Figure 6A-B: Quantification of V-ATP5A, III-UQCRC2, etc (except may be IV-COX2) in my opinion does not reflect western blot data if provided image is representative

7) Authors suggest that hnRNP U and DDX3 have a similar function in mito respiration ("discussion part"). What are effects of DDX3 depletion on OXPHOS complex protein expression? Could modest effects of hnRNP U depletion be explained by the presence of DDX3 or even compensatory effects from DDX3? Authors should 1) detect levels of DDX3 upon hnRNP U depletion; 2) determine whether DDX3/hnRNP U co-depletion will have additive or synergistic effects on mito protein translation/ expression and mito metabolism.

8) No molecular mechanisms (except binding to RNA oligos) are provided here. Showing "bind,unfold, lock) (via consecutive RG4 rearrangements and binding of hnRNP U, DDX3 and GRSF1) would make this work stellar.

9) How would RG4 in 3'UTRs regulate mRNA translation?

Version 1:

Reviewer comments:

Reviewer #1

(Remarks to the Author)

The authors addressed all comments by conducting additional experiments, rephrasing the text or explaining that their experimental attempts failed. I have one more comment. The authors use the term "mitochondrial mRNA" to describe the nuclear-encoded mRNA that encode proteins located in mitochondria. This will be confusing because "mitochondrial mRNA" is used to describe mRNAs encoded in the mitochondrial genome. The running title and the following lines need to be changed (my apologies for not pointing this out in the first round of reviews): 110, 141, 272, 274, 365, 378, 420, 422, 426/427, 514.

Reviewer #2

(Remarks to the Author)

The authors have significantly improved their manuscript and addressed my principal concerns. I note two minor points:
1. I was unable to see how they made recombinant hnRNP U used in Fig 5c, perhaps I missed it.
2. In supplementary Fig 5d the legend refers to 3' UTR data but the figure labels one of the lines as 5' UTR and none as 3' UTR. This needs sorting out. Why not show the distribution for both 5' and 3' UTR in this figure as you do in Fig 3g to unequivocally show that CBP20 distribution is different.

Reviewer #3

(Remarks to the Author)

I think that revised version improved but not in the part to describe mechanisms of localized translation and its regulation by RG4s. In my opinion it is not suitable for publication in Nature Communications

Major points:

1) It is important to show not only some partial "enrichment" of mRNAs with putative RG4s by comparison of APEX proximity data at the OMM compared to the ERM. It is actually important to demonstrate that translation of such mRNAs is indeed localized. I think AKAP1 experiments are important but without showing that AKAP1 protein is synthesized upon recruitment of AKAP1 mRNA to the OMM (e.g. by metabolic labeling and fractionation).

2) It is still not clear how putative 3'-UTR RG4s regulate translation. Will placement of any well-characterized RG4 (e.g. N-Ras) make reporter mRNAs localized and to me translated at the OMM? The puromycin test used by authors is too harsh.

3) As reviewer 1 suggested, 7-deazaG experiments are required and are more informative than mutational analysis. It is worrisome that authors failed to in vitro transcribe such 7-deazaG-containing reporters because their proteomic analysis is based on the association of proteins with 7-deazaG-containing oligos

4) Related to point 3, authors should show (by CD or other methods) that 7-deazaG-containing oligos (from in vitro

transcription) actually fail to assemble RG4s

5) Is there evidence that putative RG4s are actually folded in cells under experimental conditions? BG4-based approach is indicative but not assuring since number of things can happen during fixation-permeabilization for IF studies.

6) mTOR connection for regulation SPECIFICALLY through RG4s is weak. Studies showing that mTORC1 controls mitochondrial activity and biogenesis through 4E-BP-Dependent translational regulation (e.g., PMID: 24206664 and PMID: 28918902 are well known). How does RG4s/DDX3/nhRNP U axis fits published data?

Version 2:

Reviewer comments:

Reviewer #3

(Remarks to the Author)

In the revision, authors tried to answer on some of my questions regarding the mechanisms connecting RG4s to the crosstalk between protein synthesis and energy metabolism. Unfortunately, I do not see any improvements since initial submission and cannot recommend it for publication at such prestigious journal as Nature Communications.

1. Authors proposed that "dynamics of RG4s" is central to mRNA translation/ energetic output of certain RG4-bearing transcripts. In fact, this is not shown. Lengthy (up to few days) treatments with RG4 (and DNa G4s) ligands are too pleiotropic. I can imagine that hundreds if not thousands of transcripts will be affected and some may influence energetic balance of cell and, of course, cell death. So, how RG4 treatments mimic dynamics here??

2. 7-deazaG experiments are required! They are more informative than mutational analysis. It is worrisome that authors failed to in vitro transcribe such 7-deazaG-containing reporters because their proteomic analysis is based on the association of proteins with 7-deazaG-containing oligos. It is worrisome since authors have previously used this approach

3. Why would not authors simply place any well characterized RG4 in 3'-UTR and test whether it will affect translation?

4. I do not see rescue experiments in siRNA experiments

5. Using BG4 antibody for RNA immunoprecipitation that does not include fixation/permeabilization steps is still troublesome. Why not using SHAPE or other analysis.

6. No further mechanistic insights into RG4s/DDX3/nhRNPU are provided during revision. This is major drawbacks.

7. Authors include more unnecessary "maybes" into their work. Why would you suddenly speculate on uORFs and/or mRNA circularization if even they have not answered on previous questions on roles of RG4s in 3'UTR for mRNA translation

8. The problem of mRNA transfection of mutated vs WT RG4 is that it does not tell anything about the colocalization and localized translation at mitochondria. FISH is needed in this case (to show localization) and then some type of PLA (as an example). So, I find this experiment inconclusive

9. Finally, not all RG4s in nuclear-encoded mitochondrial mRNAs are actually found in the vicinity to mitochondria. What is the contribution of the free cytosolic versus mitochondrial fractions? Polysome analysis here is not informative

Version 3:

Reviewer comments:

Reviewer #3

(Remarks to the Author)

In this revision, authors adequately addressed my concerns. I recommend this work for publication

NCOMMS-22-33635A: *"RNA G-quadruplex dynamic steers the crosstalk between protein synthesis and energy metabolism"*

By Dumas, L, et al.

We are grateful to the reviewers for their comments and suggestions that contributed to further expanding and strengthening the study. For clarity's sake, the modifications in the revised manuscript are indicated in blue.

Detailed Response to Reviewers' Comments:

Reviewer 1

We are grateful to the Reviewer 1 for his constructive comments about the soundness of the experiments and the general interest of the manuscript.

Main comment R1:

the major conclusion that G4 formation regulates the translation of mitochondria-associated mRNAs needs direct evidence. The Authors should map G4 motifs in a few selected mRNAs, change their sequence (or synthesized using 7-deaza G) to prevent G4 formation, and test their translation after mRNA transfection into cells.

To address the RG4 requirement for mitochondrial mRNA translation, we transfected HCT116 and Hela cells with mitochondria location-specific RNA reporters containing the 3'UTR of two RG4-containing mitochondrial transcripts (GALNT2, B4GALT1; as shown in Supplementary Fig. 3g) in which (experimentally identified) RG4s were wild-type or mutated. The results showed that, for both transcripts, the RG4 mutations reduced luciferase expression in the two cell lines (Fig. 2h, Supplementary Fig. 3h). We also synthesized RNA reporters with 7-deaza G, but for some unknown reasons, their expression was very low (data not shown), making it impossible to carry out the test. GALNT-2 RNA reporters were also used to study the involvement of RG4s in OMM localization in cellulo and the role of hnRNP U in mRNA translation in vitro. We showed that mutating the RG4 at the GALNT-2 3'UTR reduced the colocalization between the mRNA reporter (Fig. 2i) and the mitochondria and prevented hnRNP U-mediated regulation of its expression (Fig. 5c) (see also R1-4). The study of the effect of the RG4 mutation on RNA-protein colocalization was hampered by the impossibility of detecting luciferase by immunofluorescence. For undetermined reasons, we were unable to detect the luciferase by immunofluorescence, which prevented us from studying the colocalization between the WT/Mut transcript and the encoded protein. These results complement those obtained with PhenDC3 and support the conclusion that RG4 formation is involved in hnRNP U-mediated mitochondria-localized mRNA translation. We made several attempts to clone AKAP, which were not successful, preventing us from carrying out reporter experiments with sequences of this transcript.

Authors describe a similar tendency for mitochondrial-encoded ND5 mRNA, which is not expected to form G4RNA.

We would like to point out that Fig. 2f does not describe the effect of the ligand on the ND5 mRNA but on the colocalization between AKAP1 and ND5 mRNAs.

Other comments:

R1-1. How the results of cell number/proliferation of HCT116 should be interpreted, seem to be contradictory (Fig.1e and Fig.S1e)

The results between the two figures are consistent: Fig. 1e shows that the effect of the ligand on proliferation are visible from 4 days onwards, which is consistent with Supplementary Fig. 1e (now Supplementary Fig. 2a) showing no effect at 24 hours of treatment. We have modified the text to clarify this point.

R1-2. Fig. 3i – cells are too small to be able to assess the staining

We have modified the figure (Fig. 3i) to zoom in on the cells.

R1-3. Why components of mitochondrial translation machinery co-IP with hnRNP U (Fig. 4f). hnRNP U is absent in the mitochondrial matrix.

Our interpretation of this result, based on previous findings on other RBP regulating translation at the OMM^{1,2}, is that hnRNP U is an OMM-localized translational player, operating in a compartment that hosts mitochondrial mRNAs as well as mitochondrial proteins to be imported into mitochondria.

R1-4. The Authors suggest that hnRNP U regulates the translation of mRNA by affecting G4RNA. This should be confirmed with a reporter mRNA that does not form G4.

As anticipated above (**Main comment R1**), we have generated mRNA reporters containing the 3'UTR of GALNT2, a RG4-containing hnRNP U target (as shown in Supplementary Fig. 3g), in which the RG4 was WT or mutated (Mut). To provide evidence of the direct involvement of hnRNP U in translational regulation, we performed in vitro translation experiments with these mRNA reporters and recombinant hnRNP U. Our results showed that the addition of hnRNP U increased luciferase expression in the presence of the WT but not with the mutated RG4 (Fig. 5c). Moreover, hnRNP U had no effect on a luciferase reporter lacking the GALNT2 3'UTR (Fig. 5c). These results support the claim that hnRNP U is a direct regulator of mRNA translation and that its activity depends on RG4-containing 3'UTRs.

R1-5. All RNAi-based experiments miss rescue controls (complementation of phenotype by RNAi-insensitive version of transgene/mRNA). Alternatively, the same phenotype should be demonstrated by CRISP-Cas9 knockout.

As requested, we performed siRNA-mediated silencing of hnRNP U and complemented hnRNP U expression by transfecting a hnRNP U plasmid in which the siRNA target site was mutated. As shown in Supplementary Figure 9b-c, transfection of this plasmid did not significantly increase hnRNP U expression under siRNA control conditions (most likely because it is a highly abundant protein) and partially rescued hnRNP U expression after siRNA treatment. Consistent with these effects, we

observed that the addition of hnRNP U did not alter mitochondrial respiration under siRNA control conditions, but partially restored it in hnRNP U depleted cells (Supplementary Figure 9d,e). Similar results were obtained when analyzing the extracellular acidification rate (Supplementary Figure 9f).

R1-6. The Authors imply that their data suggest a link between cancer and regulation of mitochondrial proteins translation. However, all experiments are done using cancer cells. Either normal, primary cells will be used for comparison, or authors will remove/tone down their claims concerning cancer biology (still, the manuscript will be interesting for being considered for publication).

To reinforce our conclusions on the biology of cancer, as an alternative to comparing HCT116 (used throughout this work) with the normal counterpart, which was not commercialized in Europe, we used MCF7 (human breast adenocarcinoma) and MCF10A (healthy breast cells) cells. We found that the effect of PhenDC3 on proliferation was stronger in MCF7 than in MCF10A. Furthermore, the ligand modified the expression of RG4-containing mitochondrial proteins targeted by hnRNP U in cancer cells, but not in normal cells (Supplementary Fig. 12c,d). These results suggest that RG4- and hnRNP U-dependent translation mechanisms may take place in cancer cells but not in normal cells. Future research will be required to provide an explanation for these differences. We have now modified the discussion to suggest that this difference offers a therapeutic opportunity.

Reviewer 2 (R2)

We are grateful to the Reviewer for the constructive comments about the potential wide interest and novelty of our work.

R2 Main comment:

- 1. HNRNPU is an extremely abundant RNA binding protein which means it appears in many mass spectrometry lists of hits. Therefore, it is not surprising that it appeared at the intersection of the datasets they used to identify proteins which might regulate RG4 translation.**

In these datasets, hnRNP U stands out when comparing different conditions. In particular, when analyzing proteins bound to a WT or mutated RG4 (as in ³) or, as we have shown previously⁴, when studying the differential binding of proteins to a folded RG4 versus a G-rich RNA unable to form RG4. Of note, our results validated these datasets by showing the preferential binding of hnRNP U to folded RG4s as compared to unfolded G-rich sequences (Fig. 3e, 4c, 5h, 7b, 7d, Supplementary Fig. 5c, 6a). Concerning the presence of hnRNP U in the translating fractions, the association of hnRNP U with the translational machinery⁵, which we validated in Fig. 5a and Supplementary Fig. 8c, is treatment-dependent (puromycin or the mTOR inhibitor). Therefore, the presence of this factor cannot be attributed solely to its abundance.

- 2. How do we know that during the 48 hour siRNA treatment the effects on these other processes, which are dramatic and well documented, do not**

indirectly alter translational processes by affecting other proteins which play a direct role in this process?

First, it's important to note that the hypothesis of a role for hnRNP U in translation was built on data showing the direct in cellulo interaction of hnRNP U with mitochondrial transcripts, which we then validated by RNA immunoprecipitation (Fig. 3d). To address more specifically the question on the possible indirect effects, which indeed cannot be ruled out given the pleiotropic role of hnRNP U in post-transcriptional regulation (shared by the majority of RBPs), we investigated whether the effects of hnRNP U on translation were direct. As described in response to **R1-4**, our results showed that addition of recombinant hnRNP U altered the expression of a luciferase reporter in a way that depended on the presence of a RG4 in the 3'UTR of GALNT-2, a mitochondrial transcript that is targeted by hnRNP U in cellulo (Fig. 5c). As a control, we showed that hnRNP U had no effect on the expression of luciferase encoded by 3'UTR-less reporter. Transcriptome analysis after hnRNP U inhibition, which we have added in the revised version (Supplementary Fig. 7f), also provided additional information to investigate the possibility of indirect effects on translation. We found that the mRNA expression of only 3 factors with a minor and poorly characterized role on translation (SH3BGRL, PDF, PIWIL4) was affected. Taken together, these arguments support a direct role for hnRNP U in mitochondrial mRNA translation via 3'UTRs.

3. Often the effects reported are small e.g. Fig. 2d

The effects of PhenDC3 and puromycin on the colocalization between the AKAP1 mRNA and the encoded protein or the ND5 mitochondrial mRNA (presently in Fig. 2d-g) are small but significant, and comparable to previous results in the presence of puromycin⁶. These data are now strengthened by additional RNAscope experiments targeting two other RG4-containing mitochondrial mRNAs, HSPD1 and GPD2, whose association with OMM was reported to be ribosome-dependent. For these two RG4-containing mRNAs, we validated the effects previously observed with puromycin using APEX-seq⁷ and extended the effects of PhenDC3 observed on AKAP1 (Fig. 2d-g) to these two mRNAs (Supplementary Fig. 3f). Of note, colocalization between RNA and protein could only be studied for HSPD1 (Supplementary Fig. 3f) due to the diffuse GPD2 immunofluorescence signal (data not shown).

R2 Specific points:

R2-1. In the abstract and introduction, the authors refer to RNAs frequently when I think they mean mRNAs. Using the term RNA is misleading because there are many other types of RNA besides mRNA. When I first read the abstract I had the impression that they were talking about a non-coding RG4 RNA regulating translation not an RG4 embedded within an mRNA.

We made "RNA to mRNA" modifications where appropriate.

R2-2. Looking at Fig 1A my impression is that the BioCyTASQ probe mainly lights up the cytoplasm where mitochondria are so without seeing the distribution of some general cytoplasmic marker not found in mitochondria as a

control it is difficult to unequivocally arrive at the conclusion that RG4s map to mitochondria.

To address this issue, we performed colocalization experiments by visualizing RG4s with BioCyTASQ and detecting by immunofluorescence either TOMM20, an OMM-localized factor, or IκB, a cytoplasmic protein not found in mitochondria. Our results showed that TOMM20 colocalized with BioCyTASQ (Fig. 1a) but not with IκB (Supplementary Fig. 1e,f), thus supporting our conclusion that RG4s mapped to mitochondria. Fig. 1a of the previous version showing the colocalization between BioCyTASQ and the mitochondrial marker MTdR has been placed in Supplementary Fig. 1c.

R2-3. Whilst PhenDC3 alters proliferation of cancer cells, is this effect specific to cancer cells? Without evidence to suggest that is the case it is not a particularly interesting result. Lots of compounds alter cell proliferation but only a more restricted group specifically alter the proliferation of cancer cells.

As indicated in response to **R1-6**, we performed proliferation experiments in breast malignant (MCF7) and normal (MCF10A) cells in the presence of PhenDC3. Our results suggest that the proliferation of normal cells is less affected by the ligand than that of cancer cells, indicating a specificity of PhenDC3 towards cancer cells (Supplementary Fig. 12c).

R2-4. In fig. 2B it appears that some RG4 containing mRNAs are translated better and some worse following stabilization of RG4s. Why the differential effects? This is not discussed and argues against a simple model whereby RG4s recruit proteins which unwind them and then drive translation of those mRNAs.

The effects of PhenDC3 on protein expression (Fig. 2b,c) are consistent with the idea that RG4s are known to primarily inhibit translation, but as previously indicated for a limited number of transcripts⁴ and shown in our results (Fig. 2b,c), they can also enhance it. This depends on the molecular mechanisms involved, and in particular on RG4s localization within UTRs/CDS and their interplay with other translation-regulating factors. The effects may also be influenced by the specific ligand used and its ability to bind RG4s and, possibly, to compete with trans-acting factor binding⁸. Given these arguments, even if a general model is often proposed, other possibilities emerge, but their study is necessary at the level of individual RG4 and transcript to be able to assert that the general model applies to specific transcripts. The revised version includes a discussion of this point.

R2-5. The data in Supp fig. 2C showing that perturbation of RG4 containing mRNAs has an impact on RG4-less mRNAs, establishes that indirect effects are at play in this system. This again reinforces how important it is to design experiments which directly rather indirectly test the hypotheses presented.

As mentioned above (**Main comment R1 and R1-4**) in the revised version, using RNA reporters in *cellulo/vitro*, we provide results directly demonstrating that the translation of mitochondrial mRNAs depends on i) the 3'UTR of mitochondrial transcripts (Fig 5c), ii) the ability of G-rich sequences to form RG4s (Fig. 2h,i, 5c, Supplementary Fig. 3h) and iii) the addition of hnRNP U (Fig 5c).

R2-6. The effects presented in Fig 2c seem very small. It's hard to be persuaded these are direct effects which I would have expected to be more dramatic.

R2-7. Fig. 2e,f. Again are these indirect effects since RG4 stabilisation no doubt has many effects on the cell? A more direct test would have been to engineer an mRNA to remove the RG4 region and demonstrate a direct effect on its mitochondrial localization and translation.

These issues have been addressed in our response to R2 Main comment 2 and 3

R2-8. Fig. 3g. Whilst this figure looks superficially compelling, what does this plot look like for other proteins. There are eCLIP datasets available for hundreds of protein in ENCODE so how HNRNPU specific is this distribution?

Analysis of the density of binding of other RBPs around RG4 elements indicated that 62/132 RBPs (47%) of the ENCODE eCLIP dataset do not have the same binding profile as hnRNP U (i.e. binding density-based ranking of transcript regions is different). We modified the text accordingly and have added a figure showing the density of an RBP (NCBP2) with a different profile with respect to that of hnRNP U (Supplementary Fig. 5d).

R2-9. HNRNPU is an essential gene and we are told that in one set of experiments its knockdown by RNA has no effect on proliferation (Supp Fig 5e) yet later in the paper we see that there is indeed an effect on proliferation at later time points. I suspect proliferation is already affected at the 48 hour time point but perhaps the cells have not yet fully detached and died. This brings me on to the point that 48 hours of HNRNPU siRNA treatment provides ample opportunity to alter splicing of many pre-mRNAs and chromatin localisation of many RNAs and a myriad of other effects e.g. alterations to mRNA stability which HNRNPU also regulates. So how do we know that indirect effects on some other mRNA are not causing the effects on translation of RG4 containing mRNAs?

These issues have been addressed in our response to R2 Main comment

R2-10. The effects of HNRNPU KD on AKAP1 mRNA translation (Fig. 5b) seem extremely small if they even exist yet this mRNA is repeatedly used as an example for follow on experiments e.g. Fig. 5d where again the effects are very modest. This data would have been more convincing if the RNA in polysomes had been subject to mRNA seq and a global view of the effects on translation presented. A single negative control mRNA ActB is presented which makes it hard to derive any statistical significance about whether RG4 containing mRNAs are preferentially altered in their translation following loss of HNRNPU.

We answered these questions by:

1. Validating translational effects upon hnRNP U silencing by performing RT-qPCR analysis of mitochondrial transcripts after pooling polysomal fractions. As shown in Fig. 5b, hnRNP U depletion induced a significant modification in mRNA translational efficiency.

2. Performing RT-qPCR analysis of a second control mRNA, GAPDH. Inhibition of hnRNP U did not alter the translation of this mRNA either (Fig. 5b).
3. Analyzing the translational regulation by hnRNP U at the transcriptome-scale by coupling polysomal analysis to RNA sequencing and validating the results by RT-qPCR analysis of mRNAs from pooled HP fractions (Supplementary Fig. 7f-h).

R2-11. Fig 3e. The affinity chromatography could be revealing an indirect interaction between hnRNPU and the RG4 via another protein. If hnRNPU selectively binds RG4s how does it do it? Via its RGG box? Some functional complementation using mutant forms of HNRNPU might have been useful to address this important mechanistic issue.

As suggested, we performed RNA affinity chromatography using cytoplasmic extracts from cells transfected with plasmids encoding hnRNP U full length (WT) or a mutated version in which the RGG domain was deleted (Δ RBD). As shown in Fig. 7b, the RGG domain was essential for hnRNP U binding to folded RG4s. To define whether this domain was also critical for the interaction between hnRNP U and its partners, we performed a co-immunoprecipitation analysis. We found that the interaction between hnRNP U and GRSF1 or DDX3X depends on the presence of the hnRNP U RGG domain (Fig. 7a). Furthermore, consistent with the proteomic analysis, hnRNP U bound directly to DDX3X, while the binding to GRSF1 was mediated by the RNA (Fig. 7a,e).

Reviewer #3 (Remarks to the Author):

We are grateful to the Reviewer for his constructive comments about the importance and originality of our work.

R3-1) Abstract suggests model that " whereby the RG4 folding dynamics, under the control of oncogenic signaling and modulated by small molecule ligands or RG4-binding proteins, modifies mitochondria-localized cytoplasmic protein synthesis". I do not think that provided data supports such model directly.

R3-2) Connection to cell signaling, especially to oncogenic signaling, is weak. Data based on Fig 5e and Sup 5f-g is not sufficient for such statement

As requested, we further explored the link between mTOR and OMM-localized translational regulation of mitochondrial transcripts by visualizing RG4s after mTOR inhibition and analyzing the colocalization between AKAP1 mRNA and the encoded protein. At first, consistent with previous results showing that RG4 folding can increase under stress conditions⁹ we showed that rapamycin (an mTOR inhibitor mimicking the effect of starvation) increased RG4 folding (Supplementary Fig. 8a). Then, combining RNA scope and immunofluorescence analysis, we showed that mTOR inhibition reduced AKAP1 mRNA/protein colocalization (Supplementary Fig. 8b), suggesting that mTOR regulates the localized translation of OMM-associated mRNAs.

R3-2) Authors have not used any rescue experiments. This is huge minus. All depletion experiments should be accompanied by rescue.

As reported in **R1-5**, we have now performed rescue experiments and observed that the addition of ectopic hnRNP U partially restored hnRNPU depletion and, proportionally, the effect of hnRNP U inhibition on mitochondrial respiration and ECAR (Supplementary Fig. 9b-9f).

R3-4) Experiments using G4 ligands are necessary but pleiotropic in nature. Experiments with PhenDC3 should be repeated using other ligand (Fig 2).

In addition to PhenDC3, the first submission included experiments with a second ligand, cPDS, showing differential mitochondrial protein expression in HEK 293 and HeLa cells (Supplementary Fig. 1b) and a modified mitochondrial respiration capacity in HeLa cells similar to that observed with PhenDC3 (now Supplementary Fig. 4b,c). In the revised manuscript, we added experiments showing that cPDS and PhenDC3 have similar effects on RG4 stabilization (visualized with the QUMA probe, given the observation that cPDS and BioCyTASQ are in competition) (Supplementary Fig. 4a). Moreover, similarly to PhenDC3, cPDS reduced the colocalization between a RG4-containing mitochondrial transcript and mitochondria (Supplementary Fig. 4d). Diffuse QUMA staining precluded the colocalization analysis between RG4s and TOMM20 as performed between BioCyTASQ and TOMM20 in Fig 1a.

R3-5) If effects of Torin 1 are very specific to the hnRNPU-mediated regulation, would inhibition of mTOR result in the similar effects as observed with puromycin and PhenDC3 (Fig 2C-f)?

As discussed in response to **R3-1,2**, in the revised manuscript we showed that, as shown for PhenDC3 and puromycin, Rapamycin inhibited the colocalization between AKAP1 protein/mRNA (Supplementary Fig. 8b).

6) Figure 6A-B: Quantification of V-ATP5A, III-UQCRC2, etc (except may be IV-COX2) in my opinion does not reflect western blot data if provided image is representative

We have repeated these experiments by changing the lysis protocol, which has produced clearer and more reproducible results (Fig. 6a,b).

7) Authors should 1) detect levels of DDX3 upon hnRNP U depletion;

We carried out the requested experiments and observed that there was no cross-regulation between the two factors (Fig. 7f).

2) determine whether DDX3/hnRNP U co-depletion will have additive or synergistic effects on mito protein translation/ expression and mito metabolism.

We carried out the requested experiments and observed that co-depletion of the two factors was additive but not synergistic on OXPHOS (Fig. 7g,h) and mitochondrial respiration (Fig. 6c,d).

8) No molecular mechanisms (except binding to RNA oligos) are provided here.

Showing "bind,unfold, lock) (via consecutive RG4 rearrangements and binding of hnRNP U, DDX3 and GRSF1) would make this work stellar.

To address this issue, we performed RNA immunoprecipitation experiments using the GRSF1 and DDX3X antibodies and cytoplasmic extracts from cells treated or not with the hnRNP U siRNA. Our results showed that DDX3X did bind to RG4-containing transcripts as hnRNP U, and that depletion of the latter did not alter their binding (Fig.7c). GRSF1 followed a similar trend (Supplementary Fig. 10e). In addition, we showed that **1)** the RGG domain was important for the interaction between hnRNP U and DDX3X (Fig. 7a) and between hnRNP U and the RG4 (Fig. 7b), **2)** the interaction between GRSF1 and hnRNP U was RNA-mediated and therefore not direct (Fig. 7e), **3)** DDX3X silencing reduced hnRNP U binding to RG4s (Supplementary Fig. 7d), suggesting that the binding of hnRNP U to the RNA is facilitated by DDX3X. Taken together, these results suggest that, as in the previously proposed "bind, unfold, lock" model⁴, hnRNP U is recruited directly by DDX3X and that, following unfolding of RG4s by DDX3X and an indirect interaction between GRSF1 and hnRNP U, GRSF1 binds RG4s and keeps them unfolded. According to this model, the depletion of DDX3X and GRSF1 should increase the folding of RG4, which we validated in Supplementary Fig. 10a.

9) How would RG4 in 3'UTRs regulate mRNA translation?

The question of how RG4s in the 3'UTR regulate mRNA translation is indeed very important and deserves to be addressed in depth. As mentioned in the discussion, one possibility emerging from recent findings is that RG4s regulate translation of downstream open reading frames (dORFs). To support this hypothesis, we analyzed published data identifying dORFs^{10,11} and showed an overlap between RG4s, dORFs and hnRNP U binding sites (Supplementary Fig.11). We believe that answering this question more thoroughly is interesting but outside the scope of this work.

References:

- 1 Gao, J. *et al.* CLUH regulates mitochondrial biogenesis by binding mRNAs of nuclear-encoded mitochondrial proteins. *J Cell Biol* **207**, 213-223, doi:10.1083/jcb.201403129 (2014).
- 2 Hemon, M., Haller, A., Chicher, J., Duchene, A. M. & Ngondo, R. P. The interactome of CLUH reveals its association to SPAG5 and its co-translational proximity to mitochondrial proteins. *BMC Biol* **20**, 13, doi:10.1186/s12915-021-01213-y (2022).
- 3 Herdy, B. *et al.* Analysis of NRAS RNA G-quadruplex binding proteins reveals DDX3X as a novel interactor of cellular G-quadruplex containing transcripts. *Nucleic acids research* **46**, 11592-11604, doi:10.1093/nar/gky861 (2018).
- 4 Herviou, P. *et al.* hnRNP H/F drive RNA G-quadruplex-mediated translation linked to genomic instability and therapy resistance in glioblastoma. *Nature communications* **11**, 2661, doi:10.1038/s41467-020-16168-x (2020).
- 5 Simsek, D. *et al.* The Mammalian Ribo-interactome Reveals Ribosome Functional Diversity and Heterogeneity. *Cell* **169**, 1051-1065 e1018, doi:10.1016/j.cell.2017.05.022 (2017).

- 6 Chouaib, R. *et al.* A Dual Protein-mRNA Localization Screen Reveals Compartmentalized Translation and Widespread Co-translational RNA Targeting. *Developmental cell* **54**, 773-791 e775, doi:10.1016/j.devcel.2020.07.010 (2020).
- 7 Fazal, F. M. *et al.* Atlas of Subcellular RNA Localization Revealed by APEX-Seq. *Cell* **178**, 473-490 e426, doi:10.1016/j.cell.2019.05.027 (2019).
- 8 Lista, M. J. *et al.* Nucleolin directly mediates Epstein-Barr virus immune evasion through binding to G-quadruplexes of EBNA1 mRNA. *Nature communications* **8**, 16043, doi:10.1038/ncomms16043 (2017).
- 9 Kharel, P. *et al.* Stress promotes RNA G-quadruplex folding in human cells. *Nature communications* **14**, 205, doi:10.1038/s41467-023-35811-x (2023).
- 10 Mudge, J. M. *et al.* Standardized annotation of translated open reading frames. *Nat Biotechnol* **40**, 994-999, doi:10.1038/s41587-022-01369-0 (2022).
- 11 Wu, Q. *et al.* Translation of small downstream ORFs enhances translation of canonical main open reading frames. *The EMBO journal* **39**, e104763, doi:10.15252/embj.2020104763 (2020).

NCOMMS-22-33635A: *"RNA G-quadruplex dynamic steers the crosstalk between protein synthesis and energy metabolism"*

By Dumas, L, et al.

We would like to thank the reviewers for their comments and suggestions, which have improved the manuscript and provided new data supporting the notion of an RG4-dependent localized translation. For the sake of clarity, changes made to the manuscript during the second revision are shown in blue.

Reviewer #1

We would like to thank the reviewer for her/his positive comments on the changes made to the first revision. We agree that the term "mitochondrial mRNAs" can be misunderstood and have therefore changed it to "nuclear-encoded mitochondrial mRNAs".

Reviewer #2

We would like to thank the reviewer for her/his positive comments on the changes made to the first revision.

1. I was unable to see how they made recombinant hnRNPU used in Fig 5c, perhaps I missed it.

The recombinant hnRNPU protein has been purchased. We have modified the material and method section by adding the product reference

2. In supplementary Fig 5d, the legend refers to 3' UTR data but the figure labels one of the lines as 5' UTR and none as 3' UTR. This needs sorting out.

As suggested, we corrected the legend indicating 5'UTR.

3. Why not show the distribution for both 5' and 3' UTR in this figure as you do in Fig 3g to unequivocally show that CBP20 distribution is different.

We apologize for the confusion. The figure displays a line for each region that has at least one binding site in the proximity of an RG4 element (-200,+200nts as shown on the x axis). In the case of NCBP2 there are no such binding sites in the 3'UTR and, consequently, the line would be at 0 density for all positions. Therefore, it was not displayed as it would not be visible. To avoid confusion, we have now modified the legend of that figure to explicitly indicate this.

Reviewer #3

We would like to thank the reviewer for her/his positive assessment of the changes made to the first revision, which addressed some of the issues raised. We would also like to thank her/him for the comments on the localized mRNA translational regulation

of nuclear-encoded mitochondrial mRNAs, which we have now addressed with additional experiments.

1. It is actually important to demonstrate that translation of such mRNAs is indeed localized.

Indeed, this is an important issue that was already addressed here by using a previously published method for studying localized translation¹ based on the analysis of RNA/protein co-localization at the OMM and puromycin sensitivity. Importantly, this technique allowed us to confirm previous findings that AKAP1 translation occurs at the mitochondria¹, and that mTOR is involved in the translation of nuclear-encoded mitochondrial mRNAs^{2,3}. As suggested, we have carried out complementary experiments based on pulse metabolic labeling of newly synthesized proteins to demonstrate that RG4s are involved in translational regulation localized to the OMM. The approach, which has been recently used to monitor nascent protein synthesis⁴ and study localized mRNA translation^{5,6}, consisted in visualizing AKAP1 neo-synthesis at the OMM by combining TOMM20 immunofluorescence with proximity ligation assay (PLA) using antibodies recognizing the AKAP N-term and puromycin incorporated into nascent peptides. The results in the new Fig. 2f and supplementary Fig. 3i revealed that AKAP1 is synthesized at the OMM, in contrast to the non-mitochondrial control protein I κ B, and that PhenDC3 impaired localized AKAP1 translation. We believe that these results, together with those demonstrating RG4 conformation-dependent RNA-protein colocalization at OMM for 3 different mitochondrial mRNAs and an ectopic reporter containing wt or mutated RG4 (Fig. 2, Fig. 5, Supplementary Fig. 3,4,8) provide a strong body of evidence for the role of RG4 in localized regulation at OMM.

2. It is still not clear how putative 3'-UTR RG4s regulate translation. Will placement of any well-characterized RG4 (e.g. N-Ras) make reporter mRNAs localized and to me translated at the OMM? The puromycin test used by authors is too harsh.

As mentioned above, we used puromycin as this is commonly used to investigate local mRNA translation¹. Concerning the role of 3'UTR RG4s in mRNA translation regulation, following previous reviewers' comments, we have hypothesized that, as in 5'UTR, small open reading frames might be involved (Supplementary Fig. 11). The answer to this question deserves to be developed in further work, in which the link with uORF and the position of an RG4 in the 3'UTR will be explored in depth, as well as regulatory mechanisms involving hnRNPU and other factors.

3. As reviewer 1 suggested, 7-deazaG experiments are required and are more informative than mutational analysis. It is worrisome that authors failed to in vitro transcribe such 7-deazaG-containing reporters because their proteomic analysis is based on the association of proteins with 7-deazaG-containing oligos.

We are aware that it would have been better to explore 7dG-containing RNAs, but despite our many attempts, we are technically limited in their synthesis.

4. Related to point 3, authors should show (by CD or other methods) that 7-deazaG-containing oligos (from in vitro transcription) actually fail to assemble RG4s

Since we were unable to synthesize RNA in the presence of 7dG, we ordered oligos containing wt or mutated RG4. Unfortunately, after several time-consuming attempts by SIGMA, they only succeeded in synthesizing mutant RNAs, preventing us from carrying out RG4 folding experiments. It is important to note that here we focused on mRNAs containing previously experimentally identified RG4s^{7,8} that we were able to consistently detect with the BG4 antibody (Fig 2a, Fig. 3h).

5. Is there evidence that putative RG4s are actually folded in cells under experimental conditions? BG4-based approach is indicative but not assuring since number of things can happen during fixation-permeabilization for IF studies.

We agree with the reviewer concerning the analysis of RG4s using BG4-mediated immunofluorescence. Here, we used the BG4 antibody for RNA immunoprecipitation that does not include fixation/permeabilization steps.

6. mTOR connection for regulation SPECIFICALLY through RG4s is weak. Studies showing that mTORC1 controls mitochondrial activity and biogenesis through 4E-BP-Dependent translational regulation (e.g., PMID: 24206664 and PMID: 28918902 are well known). How does RG4s/DDX3/hnRNP U axis fits published data?

The link with mTOR is supported by several pieces of evidence including the RG4s structuration, the translation of AKAP1, the association of hnRNP U with polysomes and RG4s, as well as the post-translational modification of hnRNP U, with different but significant effects. Even if these evidences suggest this link, extensive work needs to be done to fully answer this question and define whether 4E-BP is involved in RG4-dependent regulation of nuclear encoded mitochondrial mRNAs. Indeed, it is possible that the RG4 mechanism at the 3'UTR is different from that involving 4E-BP and 5'UTR-dependent translational regulation. However, in the hypothesis of a communication between the 5' and the 3' due to mRNA circularization, it would be possible for the two mechanisms to work in synergy for certain mRNAs. In view of these different hypotheses and some others involving DDX3X and GRSF1, we prefer not to modify the discussion as we have already toned down the involvement of mTOR and indicated that the “link between RG4s and upstream oncogenic signaling requires further investigation”.

- 1 Chouaib, R. *et al.* A Dual Protein-mRNA Localization Screen Reveals Compartmentalized Translation and Widespread Co-translational RNA Targeting. *Developmental cell* **54**, 773-791 e775, doi:10.1016/j.devcel.2020.07.010 (2020).
- 2 Morita, M. *et al.* mTORC1 controls mitochondrial activity and biogenesis through 4E-BP-dependent translational regulation. *Cell Metab* **18**, 698-711, doi:10.1016/j.cmet.2013.10.001 (2013).
- 3 Morita, M. *et al.* mTOR Controls Mitochondrial Dynamics and Cell Survival via MTFP1. *Mol Cell* **67**, 922-935 e925, doi:10.1016/j.molcel.2017.08.013 (2017).

- 4 tom Dieck, S. *et al.* Direct visualization of newly synthesized target proteins in situ. *Nature methods* **12**, 411-414, doi:10.1038/nmeth.3319 (2015).
- 5 Castro-Hernandez, R. *et al.* Conserved reduction of m(6)A RNA modifications during aging and neurodegeneration is linked to changes in synaptic transcripts. *Proceedings of the National Academy of Sciences of the United States of America* **120**, e2204933120, doi:10.1073/pnas.2204933120 (2023).
- 6 Salehi, S. *et al.* Cytosolic Ptbp2 modulates axon growth in motoneurons through axonal localization and translation of Hnrnp. *Nature communications* **14**, 4158, doi:10.1038/s41467-023-39787-6 (2023).
- 7 Guo, J. U. & Bartel, D. P. RNA G-quadruplexes are globally unfolded in eukaryotic cells and depleted in bacteria. *Science* **353**, doi:10.1126/science.aaf5371 (2016).
- 8 Kwok, C. K., Marsico, G., Sahakyan, A. B., Chambers, V. S. & Balasubramanian, S. rG4-seq reveals widespread formation of G-quadruplex structures in the human transcriptome. *Nature methods* **13**, 841-844, doi:10.1038/nmeth.3965 (2016).

We would like to thank reviewer 3 for raising questions that enabled us to provide additional experiments supporting the conclusion that RG4s regulate mitochondrial localized translation. The insertion of the new figures required modifying the distribution of few figures and moving some of them into the supplementary section. All the modifications in the text are highlighted in blue.

Q1. Authors proposed that "dynamics of RG4s" is central to mRNA translation/ energetic output of certain RG4-bearing transcripts. In fact, this is not shown. Lengthy (up to few days) treatments with RG4 (and DNa G4s) ligands are too pleiotropic. I can imagine that hundreds if not thousands of transcripts will be affected and some may influence energetic balance of cell and, of course, cell death. So, how RG4 treatments mimic dynamics here??

As explained in the discussion, we used the term "RG4 dynamics" to emphasize that for RG4s to function effectively, they must be present, and being able to unfold and refold. Here, this term is not specifically related to the treatment with ligands; rather, it was used to summarize all the effects observed, including the overall impact of ligands, RG4 mutations and hnRNP U depletion on RG4 conformation, translation regulation and energy metabolism.

Concerning the pleiotropic effect of the ligands, this question was addressed in the first round of revision as requested by Reviewer 3, i.e. using another ligand. We used cPDS and found that, similarly to PhenDC3, cPDS:

- a. modified the expression of mitochondrial proteins encoded by RG4-containing mRNAs, including factors of the respiratory chain complex (Supplementary Fig. 1b)
- b. modified mitochondrial respiration capacity (Supplementary Fig. 4b, c)
- c. induced similar effects on RG4 stabilization (Supplementary Fig. 4a)
- d. reduced the colocalization between a RG4-containing mitochondrial reporter and mitochondria (Supplementary Fig. 4d).

As with any treatment, pleiotropic effects are conceivable. Here, however, several arguments argue against this possibility: 1) similar effects using 2 different RG4 ligands, PhenDC3 and cPDS, 2) functional assay tested after short treatments (mostly 16h) that do not induce cell death (appearing after 4 days treatment (Fig. 1e), 3) Proteomic analysis after treatment with cPDS did not reveal any obvious metabolism-related pathways that might indicate a ligand effect via other pathways.

Q2. 7-deazaG experiments are required! They are more informative than mutational analysis. It is worrisome that authors failed to in vitro transcribe such 7-deazaG-containing reporters because their proteomic analysis is based on the association of proteins with 7-deazaG-containing oligos. It is worrisome since authors have previously used

During the review process we were limited in the use of 7-deazaG due to shortages of this reagent (7dG trilink N-1044, used in previous works by others (Waldron, (2018 NAR) and us (Herviou (2020 Nat Comm)) at suppliers, even production disruptions. With the analogue now available, we carried out the requested experiments to quantify the effects of RG4 modulation on mitochondrially localized translation by comparing the TOMM20-associated neo-synthesis from a luciferase reporter (GALNT2) that either forms RG4 or does not because the sequence is mutated or contains the 7-deazaG analogue. The results are described below in Q8.

We also completed Fig. 2j by adding a new panel reporting the luciferase analysis from the 7 deaza-containing transcripts. This figure shows that GALNT2 reporter expression is similarly affected by the mutation and the presence of the 7-deazaG analogue. The results support the notion of RG4 dynamics discussed above (Q1), as it indicates that in order to be functional, the RG4 must be present and capable of folding.

Q3. Why would not authors simply place any well characterized RG4 in 3'-UTR and test whether it will affect translation?

To address this issue, we need to identify a sequence that is not involved in localized translation and add an RG4. Which RNA, which RG4, at which position? which surrounding sequences? Therefore, answering this question implies not just adding an RG4 to a sequence but testing a series of parameters that are currently unknown. We believe that the answer to the question of whether a RG4 is necessary and sufficient deserves to be explored in greater depth in a future work.

Q4. I do not see rescue experiments in siRNA experiments

This question, raised by Reviewer 1 in the first round of revision, was addressed by showing that ectopic hnRNP U in cells rescued both mitochondrial respiration and glycolysis (Supplementary Fig. 9b-f).

Q5. Using BG4 antibody for RNA immunoprecipitation that does not include fixation/permeabilization steps is still troublesome. Why not using SHAPE or other analysis.

Indeed, the BG4-based RIP experiments were carried out with a cross-linking step in the cell, as in a previous article (Herviou (2020 Nat Comm)). This approach allowed us to conclude that immunoprecipitated RG4 is formed in cellulo. We have modified the text to make this information clearer.

Q6. No further mechanistic insights into RG4s/DDX3X/hnRNPU are provided during revision. This is major drawbacks.

Mechanistic insights are supported by several experiments in response to the comments of Reviewer 1 and 3 during the first round, who led us to the following conclusions:

1. the RGG domain is essential for hnRNP U binding to folded RG4s (response to Reviewer 1) (Fig. 7b)
2. the interaction between hnRNP U and GRSF1 or DDX3X depends on the presence of the hnRNP U RGG domain (response to Reviewer 1) (Fig. 7a)
3. hnRNP U binds directly to DDX3X, while the binding to GRSF1 is mediated by the RNA (response to Reviewer 1) (Fig. 7a, e)
4. there is no cross-regulation between DDX3X and hnRNPU (response to Reviewer 3) (Fig. 7f)
5. co-depletion of DDX3X and hnRNP U is additive but not synergistic on OXPHOS and mitochondrial respiration (response to Reviewer 3) (Fig 6c,d, Fig 7g,h, Supplementary Fig. 9a)
6. DDX3X and GRSF1 bind to RG4-containing transcripts as hnRNP U (response to Reviewer 3) (Fig. 7c, Supplementary Fig. 10e)
7. DDX3X silencing reduces hnRNP U binding to RG4s (response to Reviewer 3) (Fig. 7d)
8. As for hnRNPU, the depletion of DDX3X and GRSF1 increases the folding of RG4 (response to Reviewer 3) (Supplementary Fig. 10a, b)

Q7. Authors include more unnecessary "maybes" into their work. Why would you suddenly speculate on uORFs and/or mRNA circularization if even they have not answered on previous questions on roles of RG4s in 3'UTR for mRNA

The hypothesis on the involvement of uORFs answers the question of how RG4s in 3'UTRs could regulate translation, which was raised in the first round of revision. This is a new avenue arising from our analyses, and we were keen to share it with the scientific community to inspire further work on this new regulatory mechanism. Many other possibilities can be envisaged, such as circularization of RNA, miRNAs, interplay with m6A and others. We believe that testing these possibilities is a major undertaking and could be addressed in future work.

Q8. The problem of mRNA transfection of mutated vs WT RG4 is that it does not tell anything about the colocalization and localized translation at mitochondria. FISH is needed in this case (to show localization) and then some type of PLA (as an example). So, I find this experiment inconclusive

As requested, and described in Q2, to provide further evidence for the involvement of RG4s in localized translation, we used the PLA-puro technique to analyze the TOMM20-associated neo-synthesis from luciferase mRNA reporters containing the GALNT2 3'UTR forming an RG4 or not, as Gs are either

mutated or replaced by 7deazaG. We observed that when the RG4 is not formed (as either mutated or 7deazaG-containing), the translation of the reporter spread away from the mitochondria (Fig. 2k and Supplementary Fig. 3l). This result therefore complements that obtained by studying the localized translation of endogenous AKAP1/HSPD1/GPD2 in the presence/absence of the ligands. Overall, these results converge towards the conclusion that the RG4 dynamics is necessary for a RG4-containing mRNA to be translated at mitochondria. We have represented these results using box plots as in (Chouaib Dev. Cell 2020) who reported the existence of localized translation factories. For the sake of homogeneity, we have used this same representation for the other figures addressing the same message about localized translation.

Q9. Finally, not all RG4s in nuclear-encoded mitochondrial mRNAs are actually found in the vicinity to mitochondria. What is the contribution of the free cytosolic versus mitochondrial fractions? Polysome analysis here is not informative

To answer this question, we quantified the fraction of mitochondrially-associated AKAP1 mRNA colocalized with the encoded protein and that remaining outside the mitochondria. We observed that colocalized AKAP1 mRNA and protein were mainly associated with mitochondria and that addition of the ligand reduced this fraction, accompanied by a proportional increase of that in the cytoplasm (Supplementary Fig. 3g).